# RL of Thoughts: Navigating LLM Reasoning with Inference-time Reinforcement Learning

**Qianyue Hao,**[*] **Sibo Li,**[*] **Jian Yuan, Yong Li**[†]
Department of Electronic Engineering, BNRist, Tsinghua University
Beijing, China

## Abstract

Despite rapid advancements in large language models (LLMs), the token-level autoregressive nature constrains their complex reasoning capabilities. To enhance LLM reasoning, inference-time techniques, including Chain/Tree/Graph-of-Thought(s), successfully improve the performance, as they are fairly cost-effective by guiding reasoning through external logical structures without modifying LLMs' parameters. However, these manually predefined, task-agnostic frameworks are applied uniformly across diverse tasks, lacking adaptability. To improve this, we propose **RL-of-Thoughts (RLoT)**, where we train a lightweight navigator model with reinforcement learning (RL) to generate task-adaptive logical structures at inference time, enhancing LLM reasoning. Specifically, we design five basic logic blocks from the perspective of human cognition. During the reasoning process, the trained RL navigator dynamically selects the suitable logic blocks and combines them into task-specific logical structures according to problem characteristics. Experiments across multiple reasoning benchmarks (AIME, MATH, GPQA, etc.) with multiple LLMs (GPT, Llama, Qwen, and DeepSeek) illustrate that RLoT outperforms established inference-time techniques in most cases and improves up to 13.4% in challenging situations. Remarkably, with less than 3K parameters, our RL navigator is able to make sub-10B LLMs comparable to 100B-scale counterparts. Moreover, the RL navigator demonstrates strong transferability: a model trained on one specific LLM-task pair can effectively generalize to unseen LLMs and tasks. Our code is open-source at `https://github.com/tsinghua-fib-lab/RL-LLM-Reasoning`.

## 1 Introduction

Recent years have witnessed unprecedented advancements in large language models (LLMs), achieving remarkable success across diverse natural language tasks (Chang et al., 2024), including translation (Xu et al., 2024), semantic analysis (Lan et al., 2024b;a), and information retrieval (Hao et al., 2024). Despite these advancements, the inherent token-level autoregressive nature of LLMs poses a significant limitation for complex reasoning tasks (Zhao et al., 2023), such as solving mathematical problems (Ahn et al., 2024) or answering intricate questions (Zhuang et al., 2023). These tasks require sophisticated logical structures and long-term dependencies that go beyond the scope of simple sequential token prediction, leaving a considerable gap between current LLM capabilities and the demands of advanced reasoning applications.

Plentiful research has been devoted to enhancing LLM reasoning. On one hand, fine-tuning approaches attain substantial improvements on pretrained LLMs (Zhong et al., 2024; DeepSeek-AI et al., 2025; Team et al., 2025). However, these methods demand massive computational resources and large-scale datasets, being costly to implement. On the other hand, inference-time techniques, exemplified by Chain-of-Thought (Wei et al., 2022), Tree-of-Thoughts (Yao et al., 2023), and Graph-of-Thoughts (Besta et al., 2024), offer a lightweight alternative by enhancing reasoning through predefined external logical structures. While cost-effective, their logical structures rely on manual design and are task-agnostic, lacking the adaptability to diverse reasoning tasks.

---

[1]Qianyue Hao and Sibo Li contribute equally to this work.
[2]Yong Li is the corresponding author, email: `liyong07@tsinghua.edu.cn`

Addressing such limitations in inference-time techniques presents significant challenges. First, reasoning tasks span various domains, including mathematics, STEM, commonsense, etc., where tasks in each domain exhibit diverse characteristics, making it infeasible to manually design logical structures specified for each task. Second, complex reasoning tasks often require multiple steps, where the problem-solving status evolves after each step, requiring dynamic adjustments to the logical structure for subsequent reasoning. Therefore, predefined logical structures fail to adapt to the changes, limiting their effectiveness in stepwise reasoning tasks. These challenges highlight the need for more adaptive inference-time techniques to handle reasoning tasks with diversity and dynamics.

Facing these challenges, we introduce **RL-of-Thoughts (RLoT)**, a framework that leverages reinforcement learning (RL) at inference time to enhance the reasoning capabilities of LLMs. Specifically, we model long-sequence reasoning as a Markov Decision Process (MDP) and design five human cognition-inspired basic logical blocks as potential actions for decision-making. Within the MDP framework, we train an RL agent, namely the navigator model, to dynamically select and combine these blocks along the reasoning process, constructing task-specific logical structures and thereby enhancing the LLM's ability to handle complex reasoning tasks. We conduct experiments across a wide range of reasoning benchmarks, including AIME (Olympic mathematics), MATH (elementary mathematics), GPQA (STEM), StrategyQA (commonsense), etc. The results demonstrate that our RLoT design outperforms various established inference-time techniques in most cases while being compatible with multiple well-known LLMs, such as GPT, Llama, Qwen, and DeepSeek. Remarkably, our RL navigator, which contains less than 3K parameters, is able to enhance the performance of sub-10B LLMs, making them comparable to much larger LLMs with $10\times$ parameters. Moreover, the RL navigator exhibits strong transferability: a model trained with one specific LLM on one task domain can effectively generalize to unseen LLMs and tasks without fine-tuning.

In summary, the main contributions of this work include:

- We propose RL-of-Thoughts (RLoT), an inference-time technique that leverages RL to adaptively construct task-specific logical structures, enhancing LLM reasoning.

- We conduct extensive experiments to verify the effectiveness of our method to improve LLM reasoning across various tasks. Compatible with multiple widely known LLMs, RLoT outperforms established inference-time techniques by up to 13.4%.

- We demonstrate the transferability and efficiency of our method, where the trained navigator model can transfer across various LLMs and reasoning tasks. With $< 3K$ parameters, it enhances multiple sub-10B LLMs to be comparable to $10\times$ larger counterparts.

## 2 PRELIMINARIES

### 2.1 LARGE LANGUAGE MODELS (LLMS)

Large language models (LLMs) are a class of advanced neural networks characterized by parameter scales up to billions, primarily trained through next-token prediction objectives. Given a sequence $\{w_1, w_2, ..., w_{t-1}\}$, the models output $w_t$ to maximize the observation likelihood in the corpus as:

$$\prod_{t=1}^{T} P(w_t|w_1, w_2, ..., w_{t-1}).\tag{1}$$

Recent advancements in LLMs, exemplified by architectures like the GPT series (Brown et al., 2020; Kalyan, 2023; Achiam et al., 2023), the Llama family (Touvron et al., 2023; Dubey et al., 2024), etc, have demonstrated remarkable proficiency across diverse natural language understanding and generation tasks, including semantic parsing, cross-lingual translation (Zhao et al., 2023; Chang et al., 2024). Meanwhile, extensive researches integrate inference-time techniques like Chain-of-Thought (CoT) (Wei et al., 2022) and Tree-of-Thoughts (ToT) (Yao et al., 2023) to enhance the multi-step reasoning capability of LLMs. On the other hand, fine-tuning strategies leverage Outcome Reward Models (ORM) and Process Reward Models (PRM) to optimize the reasoning process through reward-guided learning (Lightman et al., 2023; Wang et al., 2024c; Luo et al., 2024). These approaches address both structural limitations of auto-regressive decoding and the challenge of maintaining logical coherence in complex tasks (Xu et al., 2025).

## 2.2 Markov Decision Process (MDP)

A Markov Decision Process (MDP) provides the core framework for sequential decision-making problems. An MDP is mathematically defined by the tuple $(\mathcal{S}, \rho, \mathcal{A}, P, R)$, where $\mathcal{S}$ is the state space, and $\rho \in \Delta(\mathcal{S})$ represents the probability distribution over initial states, with $\Delta(\mathcal{S})$ being the set of all probability distributions over $\mathcal{S}$. The action space is denoted by $\mathcal{A}$. Given a specific action taken in a particular state, the state transition probability function $P : \mathcal{S} \times \mathcal{A} \rightarrow \Delta(\mathcal{S})$ and the reward function $R : \mathcal{S} \times \mathcal{A} \rightarrow \mathbb{R}$ define the likelihood of transitioning between states and the reward associated with each action. At each time step $t$, the agent chooses an action $a_t \in \mathcal{A}$ in state $s_t \in \mathcal{S}$, receives a reward $r_t$, and transitions to the next state $s_{t+1}$. The agent's objective in an MDP is to maximize the total accumulated reward over time, which is the sum of the discounted rewards obtained at each step. The cumulative reward at time step $t$ is expressed as $G_t = \sum_{k=0}^{\infty} \gamma^k r_{t+k}$, where $\gamma$ is the discount factor that weighs the significance of future rewards.

## 3 Methods

### 3.1 Overview

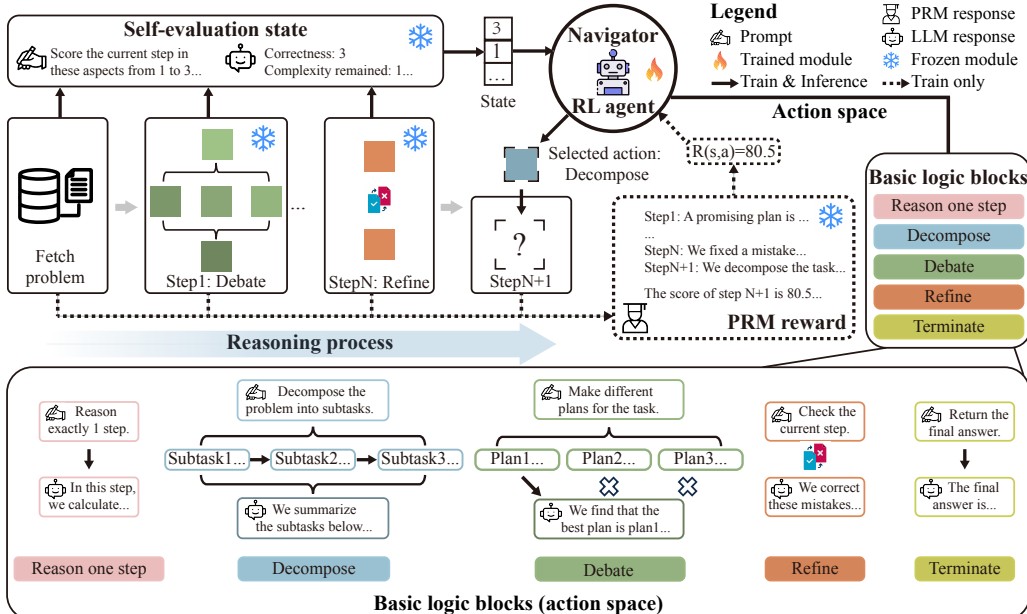

Figure 1: Framework of RL-of-Thoughts (RLoT). We train an RL agent as the navigator, which dynamically selects and combines basic logic blocks along the reasoning process, constructing task-specific logical structures for each task and thereby enhancing the LLMs' ability to handle complex reasoning tasks.

In this paper, we propose to enhance LLM reasoning at inference time with RL, namely **RL-of-Thoughts (RLoT)**. The overall framework of RLoT is illustrated in Figure 1. Specifically, we first model long-sequence reasoning as a Markov Decision Process (MDP), where we design the action space, state space, and state transition mechanism, making the sequence of decisions within an episode correspond to the generation of a logical structure for reasoning (Section 3.2). Within this MDP framework, we train an RL agent, referred to as the navigator model, which conducts sequential decision-making to construct task-specific logical structures, thereby enhancing LLMs' capability to address complex reasoning tasks (Section 3.3).

### 3.2 LLM Reasoning as MDP

To leverage the sequential decision-making capability of RL for adaptively designing logical structures at inference time based on problem characteristics, we first model long-sequence reasoning as a

specially designed MDP. Within this MDP, the sequence of decisions in an episode corresponds to the generation of a logical structure for reasoning. The designs of the state, action, reward, and state transition are as follows.

**State.** The state space is designed to capture how the current solving status of the task is after steps of reasoning, thereby supporting dynamic adjustments to the logical structure for subsequent reasoning. We employ a self-evaluation mechanism to extract concise and informative states during the reasoning process. Specifically, we prompt the LLM itself to evaluate the current reasoning steps from three major and seven detailed aspects, which are listed in Table 1.

Table 1: Aspects for state self-evaluation.

| Major | A: Correctness | B: Complexity | C: Completeness |
|---|---|---|---|
| Detailed | A1: Correctness of modeling
A2: Clarity for further reasoning
A3: Correctness of calculation | B1: Complexity to the final answer
B2: Alternative methods in
further reasoning | C1: Closeness to the final solution
C2: Completeness within the step |

For each detailed aspect, we require the LLM to assign a score from 1 to 3, which is then aggregated to form the state of the MDP. This approach summarizes complex reasoning steps into low-dimensional states, offering a comprehensive overview of the changing problem-solving status during the reasoning process, facilitating the RL agent in adjusting the strategy for subsequent reasoning accordingly. Please refer to detailed self-evaluation prompts in Appendix K.1.

**Action.** When addressing difficult and complex problems, humans often employ specific cognitive strategies. For example, we break down complex tasks into smaller components and review previous steps when encountering anomalies in the solution. As evidenced by previous research, understanding and applying these cognitive strategies can significantly enhance the reasoning capabilities of LLMs (Wu et al., 2024; Xue et al., 2024).

With inspiration from human cognition, we design five "basic logic blocks" that can be flexibly combined, constituting the action space in our MDP. By selecting and cascading the blocks, the agent thereby constructs flexible logical structures, paving reasoning pathways from the initial problem to final solutions. In detail, the basic logic blocks include:

- **Reason one step**: Perform reasoning for a single step of the current task, which may not directly lead to the final answer but contributes to the overall process.

- **Decompose**: Break the current task into simpler subtasks and execute them sequentially. Then, we prompt the LLM to briefly summarize the results of these subtasks as the final result of this action.

- **Debate**: Generate multiple plans or approaches for the task at hand and compare them to identify the most promising one. Then, we prompt the LLM to reason one step further based on the selected plan.

- **Refine**: Review and revise the current reasoning step to improve clarity and correctness.

- **Terminate**: Based on all the previous steps, provide the final answer to the original problem and show it in a specified format. This action marks the conclusion of the reasoning process.

We illustrate the structures and detailed prompts for each blocks in Figure 1 and Appendix K.2.

**Reward.** To evaluate the quality of the intermediate results after the agent selects an action during the long-sequence reasoning process, we employ the Process Reward Model (PRM) to score the intermediate results and set the PRM score of the intermediate result after each action as the single-step reward for this action.

**State Transition.** In our MDP design, the state transition is straightforward. During the reasoning process, executing a specific action based on the current problem-solving state is to prompt the LLM to continue reasoning using the logical structure corresponding to that action. After reasoning, the new problem-solving state is obtained through the aforementioned self-evaluation approach.

Also, we impose a few simple restrictions on the state transition, ensuring the correctness and rationality of the constructed logical structures. First, once the answer is already presented in the

response after executing some action, no further actions are permitted except for "Terminate". Second, the "Refine" action is automatically converted to "Reason one step" when it appears as the first action, as no refinement to the original problem is needed. Finally, to avoid the reasoning process being too long, we limit the maximum number of actions, and after reaching the limitation, the "Terminate" action will be automatically executed.

## 3.3 TRAINING OF THE NAVIGATOR MODEL

Within the MDP framework outlined above, given a specific LLM type and a kind of reasoning task, we train the navigator model. Under this formulation, the training process constitutes a standard RL problem within a discrete action space. Consequently, our framework is algorithm-agnostic, allowing for the employment of arbitrary off-the-shelf RL algorithms for training. To enhance the learning for challenging reasoning tasks, we extract hard questions from the training set of the target task, i.e., questions that the LLM cannot answer when directly prompted. Then, we use these problems for training the navigator model, from which we randomly select one in each episode and repeat it multiple times.

We illustrate the training and inference pipeline of RLoT in Figure 1. During training, we hire PRM to provide reward signals. The parameters of both the PRM and the LLM are kept fixed from pre-trained models, and only parameters of the navigator model, i.e., the RL agent, are updated. This significantly reduces the computational cost, making the training process highly efficient. After training, the PRM model is no longer required, and the trained navigator model is used directly. Given an intermediate reasoning state, the navigator selects an action, which is then used to prompt the LLM to continue reasoning using the logical structure associated with the selected action. By repeating this, the navigator model is able to guide the LLM in solving challenging reasoning tasks with task-specific logical structures.

## 4 EXPERIMENTS

### 4.1 EXPERIMENTAL SETTINGS

**Reasoning Tasks.** We conduct a comprehensive evaluation of our RLoT method across a wide range of reasoning tasks, encompassing benchmarks in mathematics, STEM, and commonsense question answering. For the mathematics domain, we adopt Olympic-level datasets, the AIME24 [1] and AMC23 [2], as well as elementary math datasets GSM8K (Cobbe et al., 2021) and MATH (Hendrycks et al., 2021c). These are widely recognized as representative benchmarks. In the STEM tasks, we test on the MMLU-STEM (Hendrycks et al., 2021b;a) and GPQA (Rein et al., 2023) datasets, which span various STEM domains and a range of difficulty levels. To evaluate the commonsense reasoning ability, we employ the StrategyQA (Geva et al., 2021) benchmark, which presents challenging multi-hop questions across diverse contexts. These benchmarks cover various domains, difficulties, and task types, forming a systematic evaluation of the reasoning ability.

**LLMs.** Our RLoT framework is designed to be independent of specific LLMs, allowing it to be compatible with any off-the-shelf LLM. To evaluate this, we test our approach using four representative LLMs: Qwen2.5-7B-Instruct, Qwen2.5-14B-Instruct (Yang et al., 2024a), Llama3.1-8B-Instruct (Dubey et al., 2024), GPT-4o-mini (Hurst et al., 2024), and DeepSeek-R1-Distill-Qwen-7B (DeepSeek-AI et al., 2025). In this paper, we mainly focus on sub-10B LLMs, which are often constrained in handling complex reasoning tasks due to their relatively smaller size. We expect that our RLoT design can substantially enhance smaller LLMs by adaptively generating task-specific logical structures at inference time, thereby making their reasoning capabilities comparable to, or even exceeding, those of much larger LLMs. Without the need to modify the LLM's parameters, this approach will be fairly computationally efficient. Note that we abandon the results of Llama3.1-8B-Instruct on the Olympic-level datasets since the capability of this base LLM is too limited to solve these challenging problems.

**Baselines.** We compare RLoT against various baselines designed to enhance LLM reasoning at inference time. First, we evaluate single-round question-answering techniques, including direct

---

[1]https://huggingface.co/datasets/AI-MO/aimo-validation-aime
[2]https://huggingface.co/datasets/AI-MO/aimo-validation-amc

question answering (Direct QA), zero-shot Chain-of-Thought (Zero-shot CoT), and few-shot Chain-of-Thought (Few-shot CoT) (Wei et al., 2022). For Zero-shot CoT, we employ prompts with "Let's think step by step", and for Few-shot CoT, we include specific few-shot examples for each benchmark in the prompts as outlined in prior work (Yang et al., 2024b; Fu et al., 2023; Rein et al., 2023; Wei et al., 2022). Additionally, we consider multi-round techniques, including self-consistent Chain-of-Thought (CoT-SC) (Wang et al., 2023) and Tree-of-Thoughts (ToT) (Yao et al., 2023). Following the original settings, we perform majority voting across four reasoning samples in CoT-SC, and we implement a logical tree with two layers and five nodes per layer in ToT. These multi-round approaches facilitate comparison and voting across diverse reasoning paths, thereby enhancing the reasoning capability of LLMs in complex tasks.

## 4.2 TRAINING DETAILS

In our implementation, the navigator model is a simple three-layer multilayer perceptron (MLP) with the Dueling Network architecture (Wang et al., 2016). The model merely contains 2,566 parameters in total, where the lightweight design ensures efficient training and inference. We employ the Double-Dueling-DQN algorithm (Mnih et al., 2015; Van Hasselt et al., 2016; Wang et al., 2016) for optimization. By integrating Double Q-learning to mitigate value overestimation and the Dueling architecture to separate state and advantage representations, this algorithm significantly improves stability during the training process. We train the navigator model for 3,000 episodes with Qwen2.5-14B-Instruct on the MATH benchmark, where the learning curves presented in Figure 2 and Appendix A.1 indicate strong convergence. Furthermore, in Appendix A.2, we investigate the impact of employing alternative RL algorithms and extending training episodes, where the results consistently demonstrate similar convergence.

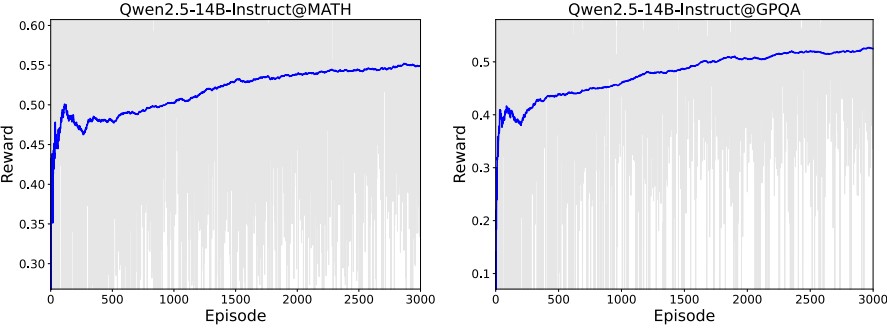

Figure 2: Learning curves during RL training of navigator models.

To obtain reward signals for actions, we utilize the Math-Shepherd as the PRM (Wang et al., 2024c). This model is trained by previous researchers using automatically constructed process-wise supervision data, reducing heavy reliance on manual annotation and thereby achieving remarkable performance. For reproducibility, please refer to more detailed settings and the illustration of the whole pipeline in Appendix J.

## 4.3 OVERALL PERFORMANCE

We use the obtained navigator model to enhance the reasoning of multiple LLMs across different reasoning tasks. Following existing works like CoT-SC, we perform multiple repeated inferences for each task and filter out the trails that do not meet self-consistency, enhancing the robustness of reasoning, where the results are presented in Table 2. For the baseline performance, we prioritize using the results reported in the official technical reports of each LLM (Yang et al., 2024a; Dubey et al., 2024; Hurst et al., 2024), if available. Otherwise, we evaluate the performance through our own experiments following the settings established in previous works (Fu et al., 2023; Yang et al., 2024b; Yao et al., 2023).

The results show that our method performs well, consistently outperforming the established inference-time baselines across almost all reasoning tasks when combined with various LLMs. Notably, our

Table 2: Overall evaluation of RLoT's capability to enhance multiple LLMs' reasoning across different tasks. The **bold** numbers indicate the best performance in each group of experiments, and the underlined numbers indicate the best baseline method.

| LLM | Method | Olympic math | | Elementary math | | STEM | | Commonsense | Average |
|---|---|---|---|---|---|---|---|---|---|
| | | AIME24 | AMC23 | MATH | GSM8K | GPQA | MMLU-STEM | StrategyQA | |
| Qwen2.5-14B-Instruct | Direct QA | 13.33 | 57.50 | 78.62 | 93.93 | 36.60 | 85.38 | 72.34 | 62.53 |
| | Zero-shot CoT | 16.67 | 62.50 | 78.56 | 94.23 | 38.39 | 85.63 | 75.98 | 64.57 |
| | Few-shot CoT | 6.67 | 55.00 | 80.00 | **94.80** | 45.50 | 85.06 | 78.60 | 63.66 |
| | CoT-SC | 6.67 | 47.50 | 80.04 | 94.08 | 45.54 | 86.71 | 80.06 | 62.94 |
| | ToT | 10.00 | 55.00 | 79.50 | 93.78 | 45.08 | 86.55 | 78.17 | 64.01 |
| | RLoT (ours) | 23.33 | 65.00 | 80.38 | 94.16 | **51.34** | 88.93 | 81.22 | 69.19 |
| Qwen2.5-7B-Instruct | Direct QA | 10.00 | 42.50 | 74.64 | 91.58 | 31.25 | 80.94 | 68.85 | 57.11 |
| | Zero-shot CoT | 16.67 | 55.00 | 74.86 | 91.58 | 34.15 | 81.00 | 71.17 | 60.63 |
| | Few-shot CoT | 13.33 | 45.00 | 75.50 | 91.60 | 36.40 | 81.16 | 74.38 | 59.62 |
| | CoT-SC | 6.67 | 50.00 | 76.36 | 92.12 | 38.84 | 83.25 | 78.45 | 60.81 |
| | ToT | 13.33 | 50.00 | 73.80 | 91.35 | 36.60 | 82.62 | 70.45 | 59.74 |
| | RLoT (ours) | 23.33 | 60.00 | 76.70 | 92.87 | 44.64 | 85.06 | 79.04 | 65.95 |
| Llama3.1-8B-Instruct | Direct QA | – | – | 49.12 | 84.83 | 32.14 | 71.58 | 70.74 | 61.68 |
| | Zero-shot CoT | | | 49.74 | 85.37 | 32.80 | 72.85 | 73.07 | 62.77 |
| | Few-shot CoT | | | 48.52 | 84.50 | 33.03 | 70.66 | 72.05 | 61.75 |
| | CoT-SC | | | 51.74 | 87.04 | 33.48 | 73.29 | 78.89 | 64.89 |
| | ToT | | | 51.93 | 86.80 | 32.24 | 74.72 | 71.47 | 63.43 |
| | RLoT (ours) | | | 56.56 | 90.07 | 46.88 | 80.56 | 84.42 | 71.70 |
| GPT-4o-mini | Direct QA | 13.33 | 55.00 | 76.58 | 93.33 | 43.08 | 85.70 | 77.00 | 61.17 |
| | Zero-shot CoT | 6.67 | 67.50 | 76.76 | **93.93** | 40.20 | 85.76 | 78.17 | 64.14 |
| | Few-shot CoT | 6.67 | 57.50 | 75.46 | 93.48 | 35.94 | 85.82 | 80.06 | 62.13 |
| | CoT-SC | 6.67 | 45.00 | 76.84 | 93.63 | 46.42 | 86.55 | 82.53 | 62.52 |
| | ToT | 6.67 | 50.00 | 74.30 | 93.33 | 44.42 | 85.92 | 76.42 | 61.58 |
| | RLoT (ours) | 20.00 | 70.00 | 77.36 | 93.86 | 54.02 | 88.23 | 82.68 | 69.45 |
| DeepSeek-R1-Distill-Qwen-7B | Direct QA | 46.67 | 60.00 | 92.27 | 95.74 | 54.47 | 83.61 | 79.48 | 73.18 |
| | Zero-shot CoT | 53.33 | 62.50 | 91.48 | 95.20 | 53.13 | 85.28 | 78.89 | 74.26 |
| | Few-shot CoT | 56.67 | 67.50 | 92.54 | 94.38 | 56.47 | 87.47 | 79.33 | 76.34 |
| | CoT-SC | 56.67 | 67.50 | 95.54 | 96.13 | 60.94 | 89.03 | 82.82 | 78.38 |
| | ToT | 50.00 | 55.00 | 95.18 | 94.54 | 55.13 | 86.55 | 80.91 | 73.90 |
| | RLoT (ours) | **63.33** | **77.50** | **96.56** | **98.94** | **67.19** | **90.77** | **86.17** | **82.92** |

approach brings about substantial improvements in the GPQA benchmark, which is a challenging task where LLMs generally perform poorly. Specifically, when implemented with Llama3.1-8B-Instruct, we achieve a 13.4% performance boost. Among the baselines, CoT-SC performs the best across most tasks. Meanwhile, we find that despite being more complex in design, ToT performs poorly on many reasoning tasks, which is also reported in previous studies (Wu et al., 2024; Zhang et al., 2024a; Qi et al., 2024). Furthermore, we report the computational overhead and implementation latency of RLoT in Appendix C, and also compare RLoT with more recently proposed test-time scaling baselines in Appendix E. By directly generate specific logical structures for each question without requiring search-and-trial, our method reached the best performance while maintain a low cost.

## 4.4 PARAMETER SIZE EFFICIENCY

In this section, we demonstrate the parameter size efficiency of our RLoT method in enhancing the reasoning capability of sub-10B LLMs, making them comparable to LLMs with several times more parameters. Specifically, we select three 10B LLMs, including Qwen2.5-14B-Instruct, Llama3.1-8B-Instruct, and GPT-4o-mini, and their respective large-scale counterparts. It is worth noting that the parameter size of GPT-4o series models was estimated in previous studies (Abacha et al., 2024).

In Appendix D, we present the performance of these models across various reasoning tasks using Few-shot CoT, which is the standard technique commonly used in official technical reports (Yang et al., 2024a; Dubey et al., 2024; Hurst et al., 2024) and previous studies (Liu et al., 2024; ret, 2024; Tran et al., 2024; Kumar et al., 2024; Yu et al., 2024) in evaluation of LLMs. We also show the performance of the 10B LLMs after enhancement with RLoT. The results indicate that our RL-based navigator, which contains fewer than 3,000 parameters, significantly enhances the performance of sub-10B LLMs, making them comparable to much larger counterparts with around 10× more parameters. Specifically, RLoT empowers the sub-10B LLMs to be comparable to, compensating most of the performance gap, or even surpassing their larger counterparts, demonstrating its efficiency.

## 4.5 TRANSFERABILITY

To better illustrate the transferability of our navigator model, we conduct further experiments regarding transferring across different LLMs and reasoning tasks, respectively.

**Transfer across Different LLMs.** To verify the transferability of RLoT across different LLMs, we respectively train navigator models with three different LLMs, namely Qwen2.5-14B-Instruct, Llama3.1-8B-Instruct, and GPT-4o-mini, on the MATH benchmark. Without any fine-tuning, we cross-test the obtained navigator models to enhance other LLMs on the MATH benchmark and present the results in Table 3.

The results indicate that the trained navigator model exhibits strong transferability across different LLMs. When implementing the navigator model to enhance the reasoning capabilities of a specific LLM, we find that, regardless of whether the navigator model is trained on the same LLM or a different one, its performance remains consistent. In all cases, the enhanced LLM outperforms the best well-known inference-time baseline.

Table 3: Evaluation of RLoT's transferability across different LLMs. We train navigator models with three different LLMs on the MATH benchmark and cross-test the obtained navigator models with other LLMs. We also list CoT-SC, the best baseline method, for comparison.

| Method | Train | Test | | |
|---|---|---|---|---|
| | | Qwen2.5-14B-Instruct @MATH | Llama3.1-8B-Instruct @MATH | GPT-4o-mini @MATH |
| RLoT (ours) | Qwen2.5-14B-Instruct@MATH | 80.38 | 56.56 | 77.36 |
| | Llama3.1-8B-Instruct@MATH | 81.48 | 53.60 | 78.14 |
| | GPT-4o-mini@MATH | 80.84 | 56.94 | 78.08 |
| CoT-SC | – | 80.04 | 51.74 | 76.84 |

**Transfer across Different Reasoning Tasks.** To verify the transferability of RLoT across different reasoning tasks, we respectively train navigator models with Qwen2.5-14B-Instruct on three different benchmarks, namely MATH, GPQA, and StrategyQA. Without any fine-tuning, We cross-test the obtained navigator models to enhance the reasoning capabilities of Qwen2.5-14B-Instruct on the other tasks and present the results in Table 4.

The results indicate that the trained navigator model owns strong transferability across different reasoning tasks. When utilizing the navigator model to enhance the reasoning capabilities of LLMs for a specific task, we observe that, regardless of whether the navigator model is trained on the same task or a different one, its performance remains largely consistent. In most cases, it surpasses the best-performing inference-time baseline.

Furthermore, we find that the navigator models trained on mathematical (MATH) and STEM (GPQA) problems exhibit better transferability to each other. However, the transferability between the navigator trained on commonsense problems (StrategyQA) and the former two is relatively limited, which is intuitive given the inherent relations and differences between domains of mathematics, STEM, and commonsense reasoning.

Table 4: Evaluation of RLoT's transferability across different tasks. We train navigator models with Qwen2.5-14B-Instruct on three different tasks and cross-test the obtained navigator models on other tasks. We also list CoT-SC, the best baseline method, for comparison.

| Method | Train | Test | | |
|---|---|---|---|---|
| | | Qwen2.5-14B-Instruct @MATH | Qwen2.5-14B-Instruct @GPQA | Qwen2.5-14B-Instruct @StrategyQA |
| RLoT (ours) | Qwen2.5-14B-Instruct@MATH | 80.38 | 51.34 | 81.22 |
| | Qwen2.5-14B-Instruct@GPQA | 80.76 | 53.57 | 80.64 |
| | Qwen2.5-14B-Instruct@StrategyQA | 79.94 | 50.22 | 81.37 |
| CoT-SC | – | 80.04 | 45.54 | 80.06 |

Table 5: Typical patterns in the task-specific logical structures generated by RLoT ("Reason" is short for "Reason one step").

| Task | MATH | GPQA | StrategyQA |
|---|---|---|---|
| Instance number | 5000 | 448 | 687 |
| Major 2-step pattern | Reason-Refine
Reason-Decompose | Reason-Refine
Reason-Debate | Reason-Debate
Reason-Refine |
| Major 3-step pattern | Decompose-Refine-Reason
Reason-Refine-Debate | Reason-Refine-Debate
Reason-Decompose-Refine | Reason-Decompose-Debate
Reason-Refine-Debate |

### 4.6 TYPICAL REASONING PATTERNS

The experimental results above have demonstrated RLoT's capability to enhance LLM reasoning with task-specific logical structures. In Table 5, we summarize typical patterns observed in the logical structures generated by RLoT when solving tasks from the MATH, GPQA, and StrategyQA benchmarks using Qwen2.5-14B-Instruct.

From the 2-step patterns, we observe that the *Reason-Refine* mode emerges frequently. Particularly in MATH and GPQA, which require massive mathematical calculations, this pattern compensates for the LLMs' relatively poor calculation abilities, leading to more reliable results. In the 3-step patterns, operations like *Decompose* and *Debate* are frequently employed, which help break down challenging problems or facilitate discussions to explore potential solutions. Additionally, the *Refine* operation is often used before and after the *Decompose* and *Debate* steps, ensuring correct integration with preceding and succeeding reasoning processes. These typical reasoning patterns exhibit strong interpretability, further validating the capability of RLoT to generate task-specific logical structures that enhance LLM reasoning. For better understanding, we provide an example in Appendix B.1 to illustrate how these logical structures empower the correct solution to a problem.

### 4.7 ABLATION STUDIES, ANALYSES, AND DISCUSSIONS

To better justify our designs, we verify the effectiveness of our logic block designs with ablation studies. To examine the impact of each block, we remove each of them and train a navigator, where results in Table 6 confirm that all blocks are effective.

Table 6: Ablation study results for each logic block.

| LLM | Ablation | MATH | GPQA | StrategyQA | Average |
|---|---|---|---|---|---|
| Qwen2.5-Instruct-7B | w/o Decompose | 75.42 | 31.92 | 77.00 | 61.45 |
| | w/o Debate | 74.02 | 36.61 | 77.58 | 62.74 |
| | w/o Refine | 75.76 | 41.29 | 72.93 | 63.33 |
| | **Full RLoT** | **76.70** | **44.64** | **79.04** | **66.79** |
| GPT-4o-mini | w/o Decompose | 76.08 | 41.74 | 79.33 | 65.72 |
| | w/o Debate | 74.68 | 45.31 | 78.75 | 66.25 |
| | w/o Refine | 76.44 | 51.56 | 74.38 | 67.46 |
| | **Full RLoT** | **77.36** | **54.02** | **82.68** | **71.35** |

Also, we conducted a series of analyses, and discussions. We quantify the contribution of each logic block in Appendix I. We discuss the reliability of the self-evaluation state design in Appendix F. We analyze the necessity of employing RL to train the navigator model in Appendix G. We illustrate the role of the process reward model (PRM) in training the navigator in Appendix H.

## 5 RELATED WORKS

### 5.1 INFERENCE-TIME REASONING OF LLMS

To improve LLMs' reasoning capability, plentiful research has investigated inference-time techniques without the need for model updates. On the one hand, predefined external logical structures are widely applied as basic solutions. Most notably, Chain-of-Thought (CoT) (Wei et al., 2022) incorporates intermediate reasoning steps within the prompt, enhancing the model's abilities in complex tasks. As a subsequent advancement, CoT with Self-Consistency (CoT-SC) (Wang et al., 2023) further refines CoT by generating multiple reasoning chains and filtering out those that do not meet self-consistency, thus increasing the reliability. Moreover, the concept of Tree-of-Thoughts (ToT) (Yao et al., 2023) and Graph-of-Thoughts (GoT) (Besta et al., 2024) has been introduced, where the reasoning process is represented as a graph, enabling exploration and backtracking from more promising outcomes. Despite their success, these methods rely on task-agnostic logical structures that are applied uniformly across diverse tasks, lacking flexibility.

On the other hand, recent researchers have proposed more adaptive inference-time approaches. From hiring the decompose-analyze-rethink procedure (Xue et al., 2024) to utilizing Monte Carlo Tree Search to discover more effective logical structures (Wu et al., 2024; Wang et al., 2024a), these approaches dynamically compose appropriate logical structures for specific tasks to enhance the reasoning performance. However, the search-and-trail design incurs massive extra computational cost, limiting the efficiency of reasoning. In contrast, the proposed RLoT method employs RL to train a navigator agent. With the trained navigator, our design can directly select and combine basic logic blocks and generate task-specific logical structures, enabling more adaptive reasoning while eliminating the searching cost.

### 5.2 RL AND LLMS

RL has become vital in the development of LLMs (Xu et al., 2025; Hao et al., 2025). One important direction is aligning LLMs with human preferences, where the key method is Reinforcement Learning from Human Feedback (RLHF) (Christiano et al., 2017; Ouyang et al., 2022). In RLHF, LLMs are fine-tuned as actors in RL based on feedback derived from human preferences. Recently, extensive helpful and harmless LLMs have been created using RLHF (Bai et al., 2022; Casper et al., 2023). Beyond this, RL is also applied to enhance the reasoning capability of LLMs. In this context, reward signals are derived from Outcome Reward Models (ORM) (Kazemnejad et al., 2024) or Process Reward Models (PRM) (Lightman et al., 2023; Wang et al., 2024c; Luo et al., 2024), providing feedbacks for the LLMs' reasoning process during fine-tuning. By applying RL to LLMs with these rewards, LLMs can iteratively improve their performance in multi-step reasoning tasks (Havrilla et al., 2024a;b; Shao et al., 2024; DeepSeek-AI et al., 2025).

In summary, existing RL techniques have significantly enhanced LLMs' capabilities by updating model parameters, yet such fine-tuning demands substantial computational resources when applied to each pre-trained LLM. In this paper, the proposed RLoT method applies RL at inference time, leveraging the power of RL to train a lightweight navigator rather than the entire parameters of LLMs, our method achieves low cost and wide compatibility with various pre-trained LLMs.

## 6 CONCLUSIONS

In this paper, we propose RL-of-Thoughts (RLoT), an inference-time technique that utilizes RL to train a navigator model, which adaptively constructs task-specific logical structures, and thereby enhances the reasoning capabilities of LLMs. Through extensive experiments across various benchmarks, we demonstrate the effectiveness of our method in improving the reasoning capability of various widely known LLMs. Additionally, we show the strong transferability and efficiency of our approach, where the trained navigator model can effectively transfer across different LLMs and unseen reasoning tasks. With fewer than 3K parameters, our navigator model enables multiple sub-10B LLMs to attain performance comparable to larger LLMs with up to 10 times the parameter size. Our work highlights the potential of RL at inference time in enhancing the reasoning capabilities of LLMs, paving the way for more adaptive and efficient LLM reasoning in the future.

## ETHICS STATEMENT

This study uses fully open-source or publicly available models and datasets, adhering to their respective licenses. All resources are properly cited in Section 1 and Section 4. The selected datasets and models are well-established, representative, and free from bias or discrimination.

## REPRODUCIBILITY STATEMENT

For Reproducibility, we describe the general experimental settings in Section 4; we list the implementation details in Appendix J; and our source code are anonymously open source at `https://github.com/tsinghua-fib-lab/RL-LLM-Reasoning`.

## ACKNOWLEDGMENTS

This work was supported in part by the National Key Research and Development Program of China under 2024YFC3307500, Tsinghua-Toyota Joint Research Institute Inter-disciplinary Program, and Beijing National Research Center for Information Science and Technology.

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

# A    TRAINING DETAILS

## A.1    LEARNING CURVES

In Figure 3, we show the learning curves during RL training of all navigator models used in experiments in the main text. We use a sliding window averaging to smooth the reward, and from the learning curves, we observe good convergence of the RL training.

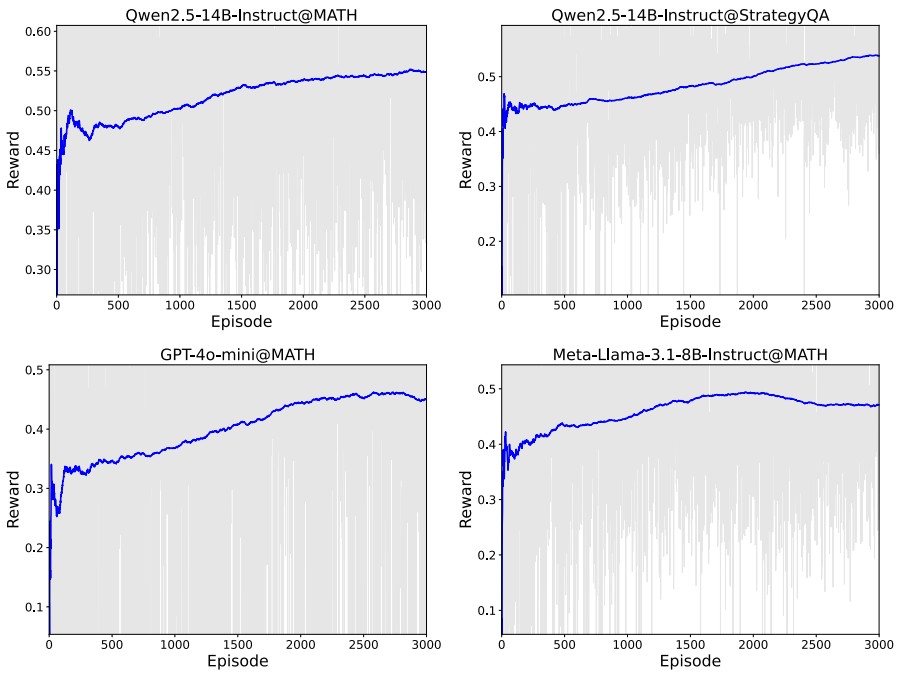

Figure 3: Learning curves during RL training of navigator models.

## A.2    ANALYSIS

In Figure 4, we extend the training duration to 5,000 episodes. We observe that the model essentially reaches convergence around 3,000 episodes, with only marginal performance gains achieved thereafter. Additionally, we explore training the navigator model using the Double-DQN algorithm. While the training stability decreases slightly due to the absence of the Dueling network architecture, the model ultimately converges to a performance level comparable to that of the Double-Dueling-DQN.

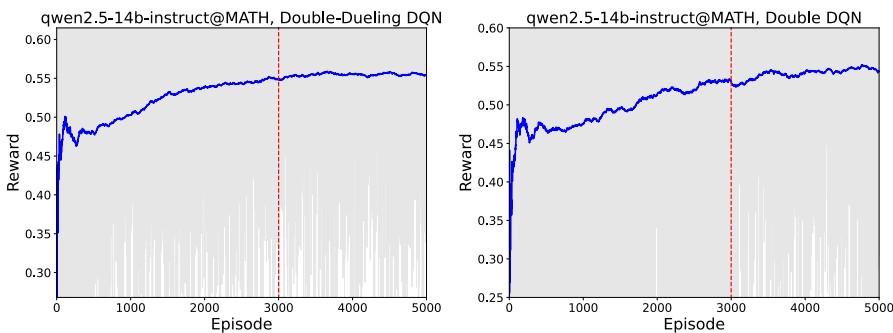

Figure 4: 5k episodes learning curves during RL training with different algorithms.

## B    CASE STUDY

### B.1    CORRECT CASES

**Problem**: How many square units are in the region satisfying the inequalities $y \ge |x|$ and $y \le -|x|+3$?

Figure 5: A case study comparing few-shot CoT and RLoT on a representative problem in the MATH dataset. Results are both generated by Qwen2.5-14B-Instruct.

As shown in Figure 5, we analyze a representative case to illustrate the detailed working procedure of RLoT. The problem is sampled from the test set of MATH (Hendrycks et al., 2021c) benchmark, which involves calculating the area between two functional curves, requiring a combination of algebraic and geometric knowledge. While the Few-shot CoT method attempts to generate the answer in one step, RLoT adopts a task-specific reasoning pathway, which includes *Reason-Refine-Debate*, a typical pattern for solving complex math problems shown in Section 4.6.

Both Few-shot CoT and RLoT successfully calculate the intersection points of the curves. However, the response of Few-shot CoT fails to accurately identify the shape of the shadowed area, leading to incorrect bases for the triangles and, ultimately, an incorrect result. In contrast, RLoT utilizes the *Refine* action to clarify the intermediate results and correctly identifies the shadowed area as a rhombus. Subsequently, the *Debate* action proposes multiple plans and prioritizes directly using the area formula for a rhombus over dividing it into triangles. By avoiding unnecessary geometric division, RLoT achieves the correct answer.

### B.2    FAILURE CASES

To provide a comprehensive understanding of the boundaries of RLoT, we also analyze typical failure cases observed during our experiments. Common failures can be categorized into two primary modes: (1) Navigator input errors caused by self-evaluation noise, and (2) Navigator output errors caused by suboptimal action selection.

As shown in Figure 6, failure happens due to self-evaluation noise. The robustness of the navigator depends partly on the accuracy of the state representation derived from the LLM self-evaluation. In this case, the LLM generates an incorrect equation (wrongly deriving $3q = 24p - 36$) but fails to identify this error during the self-evaluation phase, hallucinating a "True" status for correctness of modeling. This noise in the state input misleads the navigator: instead of selecting the necessary "Refine" action to correct the mistake, the Navigator perceives the current step as valid and selects "Debate" to explore alternative plans.

**Problem**: The polynomial $x^3 - 3x^2 + 4x - 1$ is a factor of $x^9 + px^6 + qx^3 + r.$ Enter the ordered triple $(p,q,r).$?

**State self-evaluation error**

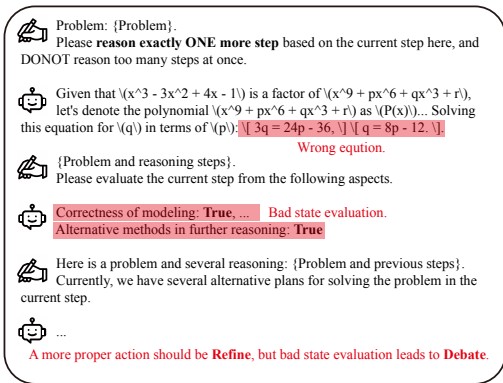

Figure 6: A failure case study that is caused by wrong state self-evaluation.

As shown in Figure 7, failure happens due to action selection error, where the navigator selects a suboptimal action despite receiving accurate state information. In this case, the LLM derives a correct intermediate result ($\cos(2\pi/7) + \cdots = -1/2$) and correctly evaluates it as "True". However, the navigator triggers a "Refine" action instead of proceeding with reasoning or termination. This unnecessary intervention forces the LLM to modify a correct step, leading to "over-correction" where valid reasoning is replaced with an erroneous identity, ultimately causing the solution to fail.

**Problem**: Compute
$$\cos^3 \frac{2 \pi}{7} + \cos^3 \frac{4 \pi}{7} + \cos^3 \frac{8 \pi}{7}.$$

**Action selection error**

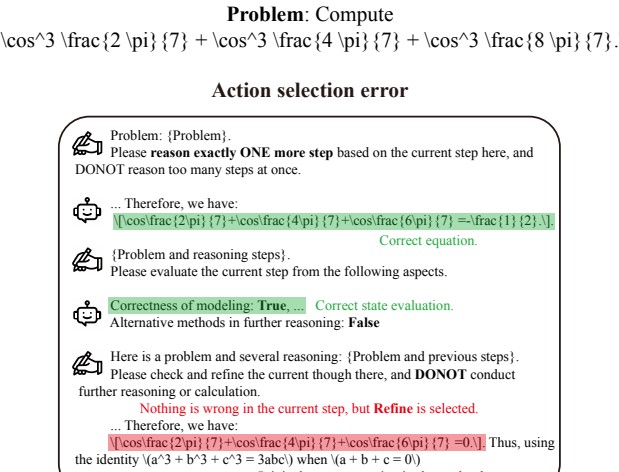

Figure 7: A failure case study that is caused by wrong action selection.

While these failure modes exist, our extensive quantitative results demonstrate that RLoT achieves performance gains over baselines, indicating that these failure cases represent a minority of the inference trajectories. In our design, the RL training process effectively optimizes the navigator to minimize action selection errors, and the structured self-evaluation prompt is designed to mitigate state noise errors as much as possible.

## C  COMPUTATIONAL OVERHEAD AND LATENCY

As shown in Table 7, RLoT improves the reasoning ability of LLMs with a low cost comparing with complex inference-time techniques. The results show that at inference-time, RLoT's token consumption only accounts for 78 % of ToT, and the consumption is comparable to CoT-SC. This is because the trained navigator can directly select appropriate logic blocks without searching like ToT, improving the performance and reducing cost. Besides, our navigator with <3k parameters has almost negligible inference cost itself.

Though RL training incurs extra costs, our lightweight navigator makes training efficient, and the trained navigator can transfer across tasks, diluting the cost. Specifically, our navigator is trained for 3000 episodes, and each episode consumes an average of 5004 input and 1448 output tokens. Shared across all test questions, it only adds less than 5% input and output tokens per question.

Table 7: Token consumption per question across tasks. The **bold** numbers indicate our method.

| Model | Method | AIME24 | | AMC23 | | MATH | | GSM8K | | GPQA | | MMLU-STEM | | StrategyQA | | Average | |
| --- | --- | --- | --- | --- | --- | --- | --- | --- | --- | --- | --- | --- | --- | --- | --- | --- | --- |
| | | Input | Output | Input | Output | Input | Output | Input | Output | Input | Output | Input | Output | Input | Output | Input | Output |
| Qwen2.5-14B-Instruct | Direct QA | 653 | 1433 | 619 | 1131 | 46 | 362 | 55 | 183 | 159 | 460 | 82 | 210 | 30 | 259 | 235 | 577 |
| | Zero-shot CoT | 667 | 1489 | 633 | 1007 | 58 | 367 | 66 | 206 | 175 | 546 | 87 | 334 | 33 | 376 | 246 | 618 |
| | Few-shot CoT | 1955 | 1341 | 1921 | 940 | 466 | 336 | 1089 | 135 | 1557 | 589 | 1114 | 259 | 421 | 223 | 1218 | 546 |
| | CoT-SC | 7823 | 5330 | 7685 | 4109 | 2014 | 1320 | 4339 | 543 | 6295 | 1945 | 4448 | 1039 | 1586 | 897 | 4884 | 2169 |
| | ToT | 5994 | 10844 | 3836 | 8029 | 4983 | 6277 | 3570 | 4312 | 6063 | 7404 | 4797 | 5363 | 5762 | 6491 | 5001 | 6960 |
| | **RLoT (ours)** | **22004** | **3888** | **17963** | **3463** | **3735** | **1109** | **2310** | **634** | **5501** | **1485** | **2923** | **791** | **2348** | **756** | **8112** | **1732** |
| Qwen2.5-7B-Instruct | Direct QA | 642 | 1714 | 644 | 1443 | 44 | 366 | 58 | 182 | 159 | 546 | 66 | 404 | 22 | 279 | 234 | 705 |
| | Zero-shot CoT | 667 | 1385 | 663 | 1180 | 47 | 393 | 78 | 222 | 184 | 522 | 82 | 413 | 32 | 401 | 250 | 645 |
| | Few-shot CoT | 4230 | 1789 | 3875 | 1501 | 483 | 581 | 1074 | 188 | 1556 | 442 | 1128 | 221 | 396 | 348 | 1820 | 724 |
| | CoT-SC | 16890 | 7131 | 15341 | 5821 | 1932 | 1443 | 4379 | 750 | 6290 | 1800 | 4433 | 883 | 1612 | 1391 | 7268 | 2746 |
| | ToT | 5289 | 9180 | 3741 | 7621 | 6679 | 7525 | 5792 | 5301 | 5771 | 6489 | 5058 | 4669 | 3807 | 2013 | 5162 | 6114 |
| | **RLoT (ours)** | **23093** | **4168** | **12876** | **2896** | **1437** | **807** | **1283** | **476** | **2112** | **889** | **1028** | **434** | **1735** | **606** | **6223** | **1468** |
| Llama3.1-8B-Instruct | Direct QA | – | – | – | – | 44 | 437 | 53 | 180 | 174 | 811 | 71 | 406 | 13 | 201 | 71 | 407 |
| | Zero-shot CoT | – | – | – | – | 73 | 495 | 64 | 215 | 175 | 661 | 91 | 365 | 20 | 418 | 85 | 431 |
| | Few-shot CoT | – | – | – | – | 510 | 472 | 1101 | 173 | 1568 | 769 | 1127 | 168 | 417 | 272 | 945 | 371 |
| | CoT-SC | – | – | – | – | 2049 | 1885 | 4325 | 808 | 6120 | 3078 | 4429 | 808 | 1582 | 1071 | 3701 | 1530 |
| | ToT | – | – | – | – | 3474 | 4307 | 5212 | 4780 | 5615 | 3794 | 5331 | 5999 | 4320 | 4868 | 4790 | 4750 |
| | **RLoT (ours)** | – | – | – | – | **4481** | **1981** | **2842** | **1028** | **9818** | **3387** | **4408** | **1486** | **3049** | **1094** | **4920** | **1795** |
| GPT-4o-mini | Direct QA | 601 | 1356 | 571 | 1184 | 43 | 404 | 54 | 223 | 165 | 397 | 86 | 175 | 13 | 201 | 219 | 563 |
| | Zero-shot CoT | 614 | 1435 | 585 | 1192 | 50 | 422 | 68 | 260 | 160 | 525 | 89 | 326 | 16 | 382 | 226 | 649 |
| | Few-shot CoT | 1853 | 1365 | 1823 | 1202 | 513 | 408 | 1107 | 205 | 1575 | 505 | 1119 | 275 | 409 | 254 | 1200 | 602 |
| | CoT-SC | 7414 | 5493 | 7294 | 5097 | 2030 | 1554 | 4442 | 823 | 6305 | 2027 | 4360 | 1005 | 1601 | 1014 | 4778 | 2430 |
| | ToT | 4639 | 4929 | 5726 | 4521 | 4534 | 5649 | 5890 | 5810 | 5175 | 5841 | 4950 | 5324 | 3833 | 1467 | 4964 | 4792 |
| | **RLoT (ours)** | **26529** | **4778** | **19513** | **3973** | **3483** | **1575** | **3069** | **1048** | **4372** | **1550** | **1882** | **710** | **1612** | **673** | **8637** | **2044** |
| DeepSeek-R1-Distill-Qwen-7B | Direct QA | 879 | 11793 | 597 | 4686 | 541 | 4960 | 533 | 1364 | 808 | 5462 | 521 | 2094 | 436 | 1268 | 616 | 4518 |
| | Zero-shot CoT | 906 | 12129 | 640 | 5140 | 565 | 4697 | 547 | 1381 | 822 | 5443 | 574 | 2012 | 450 | 1344 | 643 | 4592 |
| | Few-shot CoT | 2360 | 7778 | 2086 | 5120 | 1938 | 3301 | 3253 | 1303 | 4390 | 5194 | 2063 | 1890 | 1197 | 1275 | 2470 | 3694 |
| | CoT-SC | 9240 | 32005 | 8556 | 21859 | 7749 | 13071 | 12976 | 5123 | 17562 | 20553 | 7883 | 7820 | 4791 | 5094 | 9822 | 15075 |
| | ToT | 5879 | 34616 | 4474 | 45524 | 3076 | 25913 | 6680 | 26033 | 4481 | 19222 | 2158 | 9604 | 8393 | 11697 | 5020 | 24658 |
| | **RLoT (ours)** | **13556** | **21870** | **15684** | **31915** | **4003** | **8450** | **8347** | **12389** | **3734** | **18787** | **6732** | **8613** | **6445** | **10546** | **8357** | **16081** |
| Average | Direct QA | 694 | 4074 | 608 | 2111 | 144 | 1306 | 151 | 426 | 293 | 1535 | 165 | 658 | 103 | 442 | 308 | 1507 |
| | Zero-shot CoT | 714 | 4110 | 630 | 2130 | 159 | 1275 | 165 | 457 | 303 | 1539 | 185 | 690 | 110 | 584 | 324 | 1541 |
| | Few-shot CoT | 2600 | 3068 | 2426 | 2191 | 782 | 1020 | 1525 | 401 | 2129 | 1500 | 1310 | 563 | 568 | 474 | 1620 | 1317 |
| | CoT-SC | 10342 | 12490 | 9719 | 9222 | 3155 | 3855 | 6092 | 1609 | 8514 | 5881 | 5111 | 2311 | 2234 | 1893 | 6452 | 5323 |
| | ToT | 5450 | 14892 | 4444 | 16424 | 4549 | 9934 | 5429 | 9247 | 5421 | 8550 | 4459 | 6192 | 5223 | 5307 | 4996 | 10078 |
| | **RLoT (ours)** | **21296** | **8676** | **16509** | **10562** | **3428** | **2784** | **3570** | **3115** | **5107** | **5220** | **3395** | **2407** | **3038** | **2735** | **8049** | **5071** |
| **RLoT plus training cost** | | 21649 | 8779 | 16863 | 10664 | 3782 | 2887 | 3924 | 3217 | 5461 | 5322 | 3749 | 2509 | 3392 | 2837 | 8403 | 5174 |

Besides token consumption, we show the average solving time (min) per problem along with the standard deviation across 3 repeats in Table 8.

Table 8: Average solving time (minutes) per problem across various benchmarks. Standard deviations from 3 repeats are shown in parentheses.

| Method | AIME24 | AMC23 | MATH | GSM8K | GPQA | MMLU-STEM | StrategyQA | Average |
| --- | --- | --- | --- | --- | --- | --- | --- | --- |
| Direct QA | 0.44(0.02) | 0.25(0.01) | 0.13(0.00) | 0.05(0.00) | 0.17(0.01) | 0.08(0.00) | 0.05(0.00) | 0.17(0.01) |
| Zero-shot CoT | 0.44(0.02) | 0.25(0.02) | 0.13(0.01) | 0.06(0.00) | 0.17(0.02) | 0.08(0.00) | 0.06(0.00) | 0.17(0.01) |
| Few-shot CoT | 0.52(0.03) | 0.42(0.02) | 0.17(0.01) | 0.18(0.01) | 0.33(0.02) | 0.17(0.02) | 0.10(0.01) | 0.27(0.02) |
| CoT-SC | 2.09(0.13) | 1.74(0.12) | 0.64(0.03) | 0.71(0.07) | 1.32(0.09) | 0.68(0.05) | 0.38(0.02) | 1.08(0.07) |
| ToT | 1.86(0.07) | 1.91(0.15) | 1.33(0.12) | 1.35(0.11) | 1.28(0.06) | 0.98(0.10) | 0.97(0.11) | 1.38(0.10) |
| RLoT (ours) | 2.75(0.21) | 2.48(0.22) | 0.57(0.02) | 0.61(0.05) | 0.95(0.05) | 0.53(0.03) | 0.53(0.04) | 1.20(0.09) |

Our navigator can directly generate a task-specific logical trajectory, reducing the LLM interaction cost. Also, our lightweight navigator itself brings almost negligible cost.

# D Performance Comparison with Larger LLMs

Table 9: Performance comparison between sub-10B LLMs enhanced by RLoT, and larger LLMs with several times of parameters.

| LLM | Size | Method | MATH | GSM8K | GPQA | MMLU-STEM | StrategyQA | Average | Gap |
|---|---|---|---|---|---|---|---|---|---|
| Qwen2.5-Instruct | 14B | Few-shot CoT | 80.00 | 94.80 | 45.50 | 85.06 | 78.60 | 76.79 | -4.13 |
| | | RLoT (ours) | 80.38 | 94.16 | 51.34 | 88.93 | 81.22 | 79.21 | -1.71 |
| | 72B | Few-shot CoT | 83.10 | 95.80 | 49.00 | 89.80 | 86.90 | 80.92 | – |
| Llama3.1-Instruct | 8B | Few-shot CoT | 48.52 | 84.50 | 33.03 | 70.66 | 72.05 | 61.75 | -12.51 |
| | | RLoT (ours) | 56.56 | 90.07 | 46.88 | 80.56 | 84.42 | 71.70 | -2.56 |
| | 70B | Few-shot CoT | 68.00 | 95.10 | 46.70 | 84.81 | 76.71 | 74.26 | – |
| GPT-4o | mini (8B) | Few-shot CoT | 75.46 | 93.48 | 35.94 | 85.82 | 80.06 | 74.15 | -4.44 |
| | | RLoT (ours) | 77.36 | 93.86 | 54.02 | 88.23 | 82.68 | 79.23 | +0.64 |
| | (200B) | Few-shot CoT | 76.60 | 93.73 | 53.60 | 87.90 | 81.10 | 78.59 | – |

In table 9, we show the performance of the sub-10B LLMs after enhancement with RLoT. The results indicate that our RL-based navigator, which contains fewer than 3,000 parameters, significantly enhances the performance of sub-10B LLMs, making them comparable to much larger counterparts with around $10\times$ more parameters. Specifically, our RLoT method empowers the sub-10B LLMs to be comparable to, compensating most of the performance gap, or even surpassing their larger counterparts, demonstrating remarkable efficiency.

# E    EXTENDED BASELINE COMPARISONS

## E.1    TEST-TIME SCALING BASELINES

In addition to math reasoning models, we also compare RLoT with multiple models which cover different categories and designs. The details of baselines are listed below.
**Fixed reasoning patterns.** Baselines from this category adopts a fixed workflow to obtain the answer, which

- **Self-Refine** (Madaan et al., 2023): This method adds a simple refine step after every reasoning step to correct potential mistakes.
- **Least-to-Most** (Zhou et al., 2023): Ask LLMs to break down a complex problem into a series of simpler subproblems and then solve them in sequence.
- **SelfCheck** (Miao et al., 2024): Utilize LLMs to check their own outputs and use the results of these checks to improve question-answering performance by conducting weighted voting on multiple solutions to the question. Here we adopt voting with 4 candidates.
- **DeAR** (Xue et al., 2024): The problem is consequently decomposed, analyzed and refined in this workflow. The workflow follows a recurrent manner to a detailed decomposition of original problem.

**Tree search.** Tree-based models divide the reasoning task into sub-steps, and searches for a best path to the final answer. Tree-based methods usually requires multiple rounds Q/A, resulting in high reasoning cost.

- **One-step-greedy**: Using PRM model to greedily select the best thought at each step.
- **Basic Monto-carlo Tree Search**: An effective tree-searched method that explore the tree via multiple roll-outs. The MCTS is also supervised by the PRM.
- **Litesearch** (Wang et al., 2024a): An low-cost tree search method that assigns larger search budget to a more promising nodes.
- **$Q^*$** (Wang et al., 2024b): $Q^*$ adopts a value function that estimate the "distance" between the current thought and the correct answer. Then an $A^*$ algorithm, which is widely applied in shortest path problem, is used to finish tree-search.
- **rStar** (Qi et al., 2024): Use Monte Carlo Tree Search during inference to select actions.
- **AFlow** (Zhang et al., 2024b): An automated framework that uses MCTS to efficiently explore LLM agentic workflow.

**Others.**

- **DSPy** (Khattab et al., 2023): A programming model that can express and optimize sophisticated LM pipelines.
- **Graph of Thoughts (GoT)** (Besta et al., 2024): A graph-based model which allows arbitrary combination of thoughts according to the manually designed graph architecture. We carefully design graphs for each specific dataset.
- **Buffer of Thoughts (BoT)** (Yang et al., 2024d): A Retrieval-Augmented Generation (RAG) model that enhance reasoning via reasoning patterns in the buffer. These patterns are continuously updated during inference time.

As shown in Table 10, we compare these methods on representative datasets, GSM8K, GPQA, and StrategyQA, which covers three different domains. Results show that our model outperforms all baseline methods among all datasets.

By flexibly organizing logic blocks, our design can outperform these baselines. To compare the efficiency and effectiveness, we show the token consumption per question for representative baselines:

Unlike search-based methods that require multiple explorations, our design directly generates task-specific logical trajectories, allowing for better performance with a similar or lower cost.

Table 10: Extended baseline comparisons. The **bold** numbers indicate the best performance in each group of experiments, and the underlined numbers indicate the best baseline method. All results are averaged across 5 repeated runs with standard deviation.

| LLM | Method | | MATH-500 | GSM8K | GPQA | StrategyQA | Average |
|---|---|---|---|---|---|---|---|
| Qwen2.5-Instruct-7B | | DirectQA | 74.96±1.37 | 90.95±1.06 | 31.16±0.23 | 68.53±0.53 | 66.40±0.46 |
| | | Self-Refine | 72.32±1.20 | 88.39±0.38 | 32.23±0.61 | 74.06±0.79 | 66.75±0.33 |
| | | Least-to-Most | 73.28±0.84 | 89.10±0.82 | 31.79±0.54 | 73.60±0.64 | 66.94±0.25 |
| | | SelfCheck | 74.36±1.60 | 90.24±1.09 | 33.84±0.61 | 72.87±0.77 | 67.83±0.35 |
| | | DeAR | 69.92±0.88 | 87.84±0.54 | 35.71±0.37 | 71.03±0.85 | 66.13±0.39 |
| | Tree search | One-step-greedy | 71.24±1.43 | 89.49±0.99 | 32.63±0.50 | 72.43±0.56 | 66.45±0.54 |
| | | Basic MCTS | 72.40±0.79 | 89.51±0.72 | 35.13±0.59 | 75.72±0.80 | 68.19±0.19 |
| | | Litesearch | 72.52±1.29 | 89.86±0.86 | 33.17±0.30 | 76.04±0.65 | 67.90±0.52 |
| | | Q* | 73.76±0.81 | 92.28±0.61 | 34.38±0.76 | 76.80±0.99 | 69.30±0.50 |
| | | rStar | 75.88±1.02 | 92.05±0.89 | 38.17±0.32 | 77.09±0.83 | 70.80±0.68 |
| | | AFlow | 75.72±1.16 | 90.95±0.98 | 37.37±0.63 | 77.09±0.56 | 70.28±0.13 |
| | | DSPy | 72.48±0.86 | 88.75±0.77 | 34.11±0.68 | 73.89±0.94 | 67.31±0.49 |
| | | Graph of Thoughts (GoT) | 73.24±1.06 | 88.76±1.07 | 33.26±0.66 | 75.69±0.91 | 67.74±0.39 |
| | | Buffer of Thoughts (BoT) | 75.16±0.74 | 91.98±0.64 | 34.91±0.58 | 74.38±0.44 | 69.11±0.32 |
| | | RLoT (ours) | **76.88±1.35** | **92.95±0.63** | **44.78±0.71** | **79.56±0.64** | **73.54±0.18** |

Table 11: Token consumption, solving time (minutes) per question, and performance scores for representative baselines.

| LLM | Method | GSM8K | | | GPQA | | | StrategyQA | | | Average | | |
|---|---|---|---|---|---|---|---|---|---|---|---|---|---|
| | | Score | Token | Time | Score | Token | Time | Score | Token | Time | Score | Token | Time |
| Qwen2.5-Instruct-7B | DirectQA | 91.58 | 240 | 0.05 | 31.25 | 705 | 0.17 | 68.85 | 301 | 0.05 | 63.89 | 415 | 0.09 |
| | Self-Refine | 88.55 | 678 | 0.12 | 32.59 | 1784 | 0.41 | 73.94 | 939 | 0.16 | 65.03 | 1134 | 0.23 |
| | Litesearch | 89.76 | 912 | 0.20 | 33.04 | 2139 | 0.55 | 75.84 | 1228 | 0.24 | 66.21 | 1426 | 0.33 |
| | rStar | 91.96 | 2636 | 0.77 | 38.62 | 4127 | 1.44 | 77.15 | 2987 | 0.82 | 69.24 | 3483 | 1.01 |
| | RLoT (ours) | 92.87 | 1759 | 0.61 | 44.64 | 3001 | 0.95 | 79.04 | 2341 | 0.53 | 72.18 | 2367 | 0.70 |

## E.2 MATH DOMAIN BASELINES

To better demonstrate the effectiveness of our design, we compare RLoT with more baselines that have been specifically designed for math reasoning tasks. We involve fine-tuning designs which are computationally expensive.

- **AceMath** (Liu et al., 2024): A supervised fine-tuning model designed specifically for mathematical reasoning. It first develops a math-specialized reward model using public datasets and then performs fine-tuning and reasoning guided by this reward model.
- **PFPO** (Jiao et al., 2024): A supervised fine-tuning approach guided by pseudo reward feedback. The feedback is generated either through a self-consistency mechanism or with the assistance of more powerful LLMs.

We also include inference-time designs that hire complicated algorithms like Monte-Carlo Tree Search (MCTS).

- **LLM2** (Yang et al., 2024c): A lightweight model aimed at enhancing LLM reasoning during inference. It achieves this by training a verifier to distinguish and prioritize better LLM-generated responses.
- **LLaMA-Berry** (Zhang et al., 2024a): An inference-time Monte Carlo Tree Search (MCTS) method that explores reasoning paths using a trained reward model.
- **HiAR-ICL** (Wu et al., 2024): A method that matches problems with multiple reasoning templates at inference time. These templates are previously generated using MCTS on a subset of the dataset.

Table 12: Performance comparison of RLoT's with more baselines. The **bold** numbers indicate the best performance in each category.

| Category | Method | MATH | GSM8K | Average |
|----------|--------|------|-------|---------|
| Fine-tuning | AceMath (Liu et al., 2024) | **64.42** | **90.45** | **77.44** |
| | PFPO (Jiao et al., 2024) | 57.80 | 89.60 | 73.70 |
| Inference-time | LLM2 (Yang et al., 2024c) | 48.60 | 88.00 | 68.30 |
| | LLaMA-Berry (Zhang et al., 2024a) | 54.80 | 89.80 | 72.30 |
| | HiAR-ICL (Wu et al., 2024) | 55.00 | **90.70** | 72.85 |
| | RLoT (ours) | **56.56** | 90.07 | **73.32** |

We test with Llama3.1-8B-Insturct on MATH and GSM8K benchmarks and show the results in Table 12. For the baselines, we use the performance reported in the original papers, and for RLoT, we test with the same navigator model as in Section 4.3 in the main text. From the results, we can observe that our method outperforms all inference-time baselines. Meanwhile, with substantially lower computational consumption, it reaches comparable performance to some fine-tuning methods that require modification on the parameters of LLMs.

## F    ANALYSES ON THE SELF-EVALUATION STATE

### F.1    RELIABILITY

Previous work shows that LLMs can evaluate and correct their own outputs (Han et al., 2024). Meanwhile, multiple works (Madaan et al., 2023; Xue et al., 2024) have reached success with self-correcting. Based on the literature, we design the structured self-evaluation that prompts the LLM across granular aspects (Table 1), including the correctness of modeling and calculation. This design aims to be more specific and reliable than the single judgment.

To validate the robustness, we randomly sample 100 intermediate reasoning texts and manually check the self-evaluation. We find that the LLM's assessment was accurate in 82 out of 100 cases. Also, we deliberately introduced modeling and calculation errors in a reasoning path, and test whether our self-evaluation mechanism is able to correctly identify. As shown in Table 13, the problem is from GSM8K (Cobbe et al., 2021), which includes the relationship of three variables. In addition to the correct reasoning step, we modify it to obtain two wrong reasoning steps, which respectively makes a mistake on modeling and calculation. The result shows that the LLM is able to identify and classify different kinds of mistakes, demonstrating the effectiveness of self-evaluation state extraction.

Table 13: An example of self-assessing of intermediate states.

| Question | Reasoning step | Self-evaluated correctness | |
| --- | --- | --- | --- |
| | | Modeling | Calculation |
| Mark has a garden with flowers. He planted plants of three different colors in it. Ten of them are yellow, and there are 80% more of those in purple. There are only 25% as many green flowers as there are yellow and purple flowers. How many flowers does Mark have in his garden? | In this step, we aim at calculate the number of purple flowers, which is 80% more than the yellow ones. We calculate by $10 \times (1 + 0.8) = 18$ | True | True |
| | In this step, we aim at calculate the number of purple flowers, which is 80% more than the yellow ones. We calculate by $10 \times 0.8 = 8$ | False | True |
| | In this step, we aim at calculate the number of purple flowers, which is 80% more than the yellow ones. We calculate by $10 \times (1 + 0.8) = 17$ | True | False |

Based on the validation, we consider our structured self-evaluation provides a sufficiently reliable state representation for training the navigator.

### F.2    NOISE IMPACT

To investigate the navigator's reliance on accurate state perceptions, we introduced synthetic noise into the self-evaluation state vectors during inference. As shown in Table 14, replacing the state with random values results in a significant performance drop, falling even below DirectQA. This confirms that the navigator actively utilizes the state information for decision-making rather than following a fixed, state-agnostic policy. However, the framework exhibits commendable robustness. Under extreme conditions with 50% noise (where half of the state bits are flipped), RLoT still maintains an average accuracy of 64.99%, being comparable to the DirectQA baseline. As the noise decreases to 30%, performance recovers rapidly to 69.11%.

Meanwhile, to minimize potential noise inherent in self-evaluation, we deliberately designed the evaluation criteria as a binary classification task rather than a multi-grade continuous scoring. Continuous or fine-grained scoring (e.g., 1-10 scales) requires precise calibration, which is challenging for LLMs. In contrast, a binary standard simplifies the decision boundary, significantly reducing the cognitive load on the LLM. This design choice makes the evaluation process more robust, ensuring that the state vector remains a stable and reliable signal for the navigator even with less capable LLMs.

Table 14: Impact of noise in self-evaluation states.

| LLM | State | GSM8K | GPQA | StrategyQA | Average |
|---|---|---|---|---|---|
| Qwen2.5-Instruct-7B | DirectQA | 91.58 | 31.25 | 68.85 | 63.89 |
| | Random | 89.76 | 28.13 | 65.50 | 61.13 |
| | 50% noise | 91.74 | 34.82 | 68.41 | 64.99 |
| | 30% noise | 92.27 | 43.75 | 71.32 | 69.11 |
| | **Normal** | **92.87** | **44.64** | **79.04** | **72.18** |

### F.3 ALTERNATIVE DESIGNS

Furthermore, we explore alternative designs of self-evaluation:

- **MLP.** Use the self-evaluation vector from a fixed LLM as input, only train an MLP as the navigator.

- **LLM (fixed).** Directly prompt the LLM to select the action based on the raw reasoning trajectory.

- **LLM (finetune).** Use the raw reasoning trajectory as input, train an LLM backbone with an MLP classification head as the navigator.

Table 15: Comparison of alternative self-evaluation designs.

| LLM | Navigator | GSM8K | GPQA | StrategyQA | Average |
|---|---|---|---|---|---|
| Qwen2.5-Instruct-7B | DirectQA | 91.58 | 31.25 | 68.85 | 63.89 |
| | LLM (fixed) | 89.16 | 37.95 | 74.38 | 67.16 |
| | LLM (finetune) | 88.32 | 34.15 | 70.01 | 64.16 |
| | MLP (ours) | 92.87 | 44.64 | 79.04 | 72.18 |

As the results in Table 15 show, LLM (fixed) performs badly since pre-trained LLMs are not specified for the task. LLM (finetune) even performs worse. Our method injects valuable human knowledge for explicit, structured self-evaluation. In contrast, LLM (finetune) requires an implicit evaluation from raw text, which is much more complex and has more parameters to be tuned. This requires substantially more consumption and specialized design to overcome the complexity and uncertainty. Therefore, our method is a comprehensive design at a low cost. While some self-evaluation cases may be inaccurate, the overall improvement demonstrates the general benefit.

## G    ANALYSES ON THE ROLE OF RL IN TRAINING THE NAVIGATOR

To illustrate the necessity of using RL in training the navigator, we compare the RL navigator with several other decision methods for deciding the next reasoning block. The methods includes the

- **Fixed logic sequence.** Fixed "Decompose-Reason-Refine" sequence like human thinking.
- **Supervised-trained navigator.** We first collected 20K state-action-score samples with random actions. Then, we trained a model to predict the outcome for each action given a state, and select the action with the highest predicted score during inference.
- **LLM as navigator.** Directly prompt another LLM to select the action based on the reasoning context.

We also add:

- **Repeated strong blocks.** Repeat Debate or Refine until reaching the answer.

Table 16: Comparison with alternative navigator methods on various benchmarks.

| LLM | Method | GSM8K | GPQA | StrategyQA | Average |
|---|---|---|---|---|---|
| Qwen2.5-Instruct-7B | DirectQA | 91.58 | 31.25 | 68.85 | 63.89 |
| | Repeated refine | 87.72 | 32.37 | 72.34 | 64.14 |
| | Repeated debate | 89.99 | 33.04 | 72.05 | 65.03 |
| | Fixed logic sequence | 88.38 | 36.16 | 71.13 | 65.22 |
| | Supervise-trained navigator | 89.84 | 36.83 | 70.31 | 65.66 |
| | LLM as navigator | 89.16 | 37.95 | 74.38 | 67.16 |
| | **RL-trained navigator (ours)** | **92.87** | **44.64** | **79.04** | **72.18** |

The results in Table 16 show that these methods are worse than RLoT:

- **Fixed logic sequence.** Rigid, unable to design flexible logical structures for specific tasks.
- **Supervise-trained navigator.** The randomly sampled state-action pairs lack the ability of RL to purposely exploit effective parts of the sample space.
- **LLM as navigator.** Pre-trained LLMs are not specified for the logic selection task.

These show the importance of using RL for our navigator to flexibly select logic blocks.

# H ANALYSES ON THE PROCESS REWARD MODEL (PRM)

## H.1 THE ROLE OF PRM

To investigation the role of PRM, we also test an ORM to train the navigator.

Table 17: Comparison of RLoT using an Outcome Reward Model (ORM) vs. a Process Reward Model (PRM).

| LLM | Method | GSM8K | GPQA | StrategyQA | Average |
|---|---|---|---|---|---|
| Qwen2.5-Instruct-7B | DirectQA | 91.58 | 31.25 | 68.85 | 63.89 |
| | RLoT with ORM | 91.96 | 41.52 | 73.94 | 69.14 |
| | **RLoT with PRM (ours)** | **92.87** | **44.64** | **79.04** | **72.18** |

The results in Table 17 show that the PRM outperforms the ORM, indicating that the step reward signal from PRM is crucial.

## H.2 GENERALIZATION CAPABILITY OF PRM

Recent studies suggest that process reward models (PRMs) trained on mathematical data exhibit strong transferability to related domains such as STEM (Zhang et al., 2025). To empirically verify this and justify our reward design, we conducted a PRM calibration analysis. We calculated the correlation between the intermediate PRM scores assigned at various reasoning steps and the correctness of the final answer.

Table 18: PRM calibration.

| LLM | Correlation with Correctness | MATH | GPQA | StrategyQA |
|---|---|---|---|---|
| Qwen2.5-Instruct-7B | Step1 PRM | 0.199 | 0.184 | 0.157 |
| | Step2 PRM | 0.222 | 0.213 | 0.195 |
| | Step3 PRM | 0.236 | 0.215 | 0.202 |
| | Step4 PRM | 0.240 | 0.228 | 0.218 |
| | Average | 0.224 | 0.210 | 0.193 |

As presented in Table 18, the results reveal two insights. First, there is a consistent positive correlation between intermediate PRM scores and the final outcome across all tested benchmarks. This confirms that the PRM effectively gauges the quality of intermediate steps, validating its suitability as a dense reward signal for training the navigator. Second, although the PRM was trained primarily on mathematical data, its scores exhibit a notable correlation with solution correctness on out-of-domain tasks, including GPQA (STEM) and StrategyQA (commonsense). This demonstrates the PRM's intrinsic ability to generalize its verification logic across different domains.

This calibration analysis explains the underlying mechanism of our method's transferability. Because the math-trained PRM provides reliable feedback even on unseen tasks like GPQA and StrategyQA, the navigator trained with these rewards can successfully learn effective reasoning policies across diverse domains.

## H.3 IMPACT OF PRM QUALITY

Further, to explore the impact of PRM quality, we test:

- ORM: As a minimal quality PRM.
- Degraded PRM: Add a std=0.1 Gaussian noise to our raw PRM.
- Better PRM: Qwen2.5-Math-PRM-7B, a stronger PRM.

These results in Table 19 show that a higher-quality PRM is beneficial. However, the navigator is robustly beneficial with a lower-quality PRM, as long as the reward signal is directionally meaningful.

Table 19: Impact of PRM quality on navigator performance.

| LLM | PRM | GSM8K | GPQA | StrategyQA | Average |
|---|---|---|---|---|---|
| Qwen2.5-Instruct-7B | None-DirectQA | 91.58 | 31.25 | 68.85 | 63.89 |
| | ORM | 91.96 | 41.52 | 73.94 | 69.14 |
| | MathShepherd-Mistral-7B + disturb | 92.49 | 42.41 | 77.15 | 70.68 |
| | MathShepherd-Mistral-7B | 92.87 | 44.64 | 79.04 | 72.18 |
| | Qwen2.5-Math-PRM-7B | 93.10 | 45.31 | 80.06 | 72.82 |

# I    ANALYSES ON THE DESIGN OF LOGIC BLOCKS

## I.1    EFFECT OF LOGIC BLOCK NUMBER

We first check the test-time scaling for the logic blocks. Results in Figure 8 show that a longer sequence of logic blocks brings a performance gain.

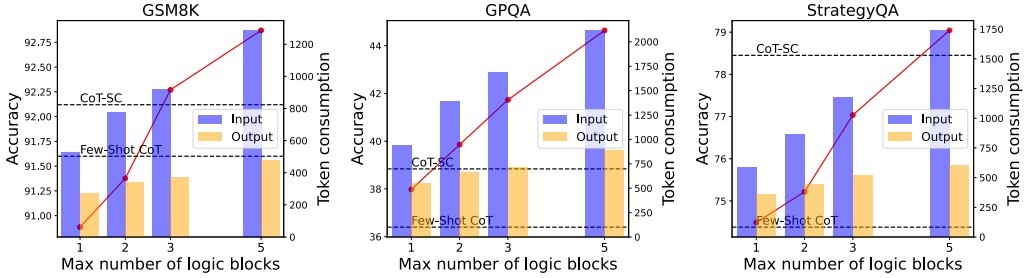

Figure 8: Test-time scaling with different number of logic blocks.

## I.2    ROLE OF DIFFERENT LOGIC BLOCKS ON DIFFERENT DATASETS

We also analyze the impact of different logic blocks on different datasets. Results in Figure 9 show the average PRM reward gain of each logic block. The most useful block varies in different tasks, specifically Debate for math tasks, Decompose for STEM tasks, and refine for common-sense reasoning.

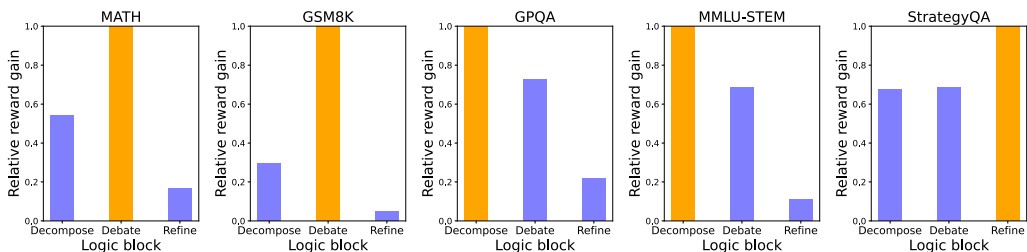

Figure 9: Contribution of each logic block in the reasoning sequence.

## I.3    ALTERNATIVE BLOCKS

Further, we test some possible new blocks:

- **Follow-up-question**: Propose and answer a sub-question based on the current reasoning.
- **Rephrasing**: Reorganize the current reasoning.
- **Compression**: Condense the current reasoning.

Table 20: Performance with additional logic blocks.

| LLM | Logic blocks | MATH | GPQA | StrategyQA | Average |
|---|---|---|---|---|---|
| Qwen2.5-Instruct-7B | Original in RLoT | 76.70 | 44.64 | 79.04 | 66.79 |
| | Original + Follow-up-question | 76.78 | 43.97 | 79.24 | 66.66 |
| | Original + Rephrasing | 77.30 | 45.54 | 80.20 | 67.68 |
| | Original + Compression | 76.06 | 43.53 | 77.87 | 65.82 |

The results in Table 20 showed that "Rephrasing" is beneficial, while "Follow-up-question" (perhaps self-ask-and-answer has limited effect) and "Compression" (likely due to information loss) are not. This shows the extensibility of RLoT: it provides a framework for the community to design more reasoning blocks to enhance the performance.

## J  IMPLEMENTATION DETAILS

### J.1  SETTINGS

In this section, we provide the main implementation settings for reproducibility in Table 21. Please refer to our source code at `https://anonymous.4open.science/r/RL-LLM-Reasoning-1A30` for the exact usage of each hyper-parameters and more details.

Table 21: Implementation details.

| Module | Element | Detail |
|---|---|---|
| System | OS | Ubuntu 22.04.2 |
| | CUDA | 11.7 |
| | Python | 3.11.4 |
| Double-Dueling DQN | $\gamma$ | 0.9 |
| | Number of episodes | 3000 |
| | Batch Size | 64 |
| | Interval of target network updating | 50 |
| | Optimizer | Adam |
| | Learning rate | 0.01 |
| | Learning rate decay | 0.5 per 1000 episodes |
| | Replay buffer size | 500 |
| | Start epsilon | 1 |
| | Min epsilon | 0.0 |
| | Epsilon decay | 0.9995 per step |
| RLoT framework | Maximum number of actions | 5 |
| | Number of trails for self-consistency | 3 |
| | Temperature for LLMs | 1.0 |

### J.2  ILLUSTRATION OF THE FULL PIPELINE

Here we illustrate how every parts in our design work together as an whole pipeline.

### 1  DATA CONSTRUCTION

We pick out "hard problems", namely those that the LLM wrongly answers when directly prompted, for training the navigator. Here are more details: we used Qwen2.5-14B-Instruct as a reference model to select "hard problems" from the training sets. The direct prompts are:

```
Prompts:
{Problem} Please generate the answer for the problem.
(MATH) Wrap the answer with boxed{{answer}}.
(GPQA) End the answer with 'The answer is (CHOICE)'.
(StrategyQA) End the answer with 'YES/NO'.
```

The statistics on benchmarks involved in our training are shown in Table 22.

Table 22: Statistics on benchmarks involved in our training.

| Dataset | Total | Hard | Proportion (%) |
|---|---|---|---|
| MATH | 7500 | 1736 | 23.15 |
| GPQA | 448 | 255 | 56.92 |
| StrategyQA | 1600 | 341 | 21.31 |

By selecting "hard problems", our navigator dedicates to strategies that enhance LLMs on problems that they cannot solve directly. For our RL approach, we do not construct (input, expected_output) pairs like supervised learning. The training data consists of individual questions. During training, the model is fed with questions, and the outputs are evaluated to generate rewards, which are then fed back to train the model. We will detail the key components and entire process of training below.

## 2 STATE SELF-EVALUATION (STATE SPACE)

We obtain the 7-aspect state vector through a prompted self-evaluation mechanism. We feed the problem and the existing reasoning into the LLM with the prompt in Appendix K.1. This design compresses the textual reasoning trajectory into a low-dimensional state vector, making the policy learning process more efficient and tractable.

Here is an input-output example of state self-evaluation, as seen in our case study in Appendix F:

```
Input
(Problem) Mark has a garden with flowers. He planted plants of three different
colors in it. Ten of them are yellow, and there are 80% more of those in
purple. There are only 25% as many green flowers as there are yellow and
purple flowers. How many flowers does Mark have in his garden?
(Existing reasoning) In this step, we aim at calculating the number of purple
flowers, which is 80% more than the yellow ones. We calculate by 10 * (1 + 0.8) = 17

Output
(State vector) {Modelling: True, Calculation: False, ...}
```

## 3 LOGICAL BLOCKS (ACTION SPACE)

**First**, these logic blocks are inspired by well-established cognitive strategies in human problem solving. "Decompose," "Debate," and "Refine" are the core actions, supplemented by "Reason one step" and "Terminate" to ensure complete reasoning flows.

For instance, "Decompose" reflects the widely studied divide-and-conquer approach, while "Debate" aligns with strategies involving comparative evaluation (Wang & Chiew, 2010). Also, they are empirically effective. Prior studies (Madaan et al., 2023; Xue et al., 2024) have demonstrated that "Decompose" and "Refine" can significantly enhance the performance of LLMs on complex tasks.

**Second**, while preserving the high-level cognitive intuition, we have operationalized them into specific primitives that the LLM can execute. The key differences are:

- **Formalization with prompts**: Each block corresponds to a prompt template that instructs the LLM to perform a specific function (Appendix K.2).
- **Standard Input/Output**: All blocks operate on a unified interface. The input is the original problem and the existing reasoning, and the output is a new segment of reasoning. This allows flexible composition of logic blocks.

**Third**, here is an input-output example for the 'Debate' block (as Appendix B.1):

```
Input
(Problem) How many square units are in the region satisfying the inequalities
$y \ge |x|$ and $y \le -|x|+3$?
(Existing reasoning) Let's start by analyzing the given inequalities. The next
step is to determine the points of intersection... Now we know the points of
intersection are...

Output
(New reasoning) The most promising plan is: Given the diagonals of the rhombus,
we can now calculate the area.
```

The contributions of the logic blocks are discussed in Appendix I.

## 4 PROCESS REWARD

The PRM takes in the problem and existing reasoning and outputs a numerical score evaluating its quality.

Here is an input-output example for the PRM:

```
Input:
(Problem) How many square units are in the region satisfying the inequalities
$y \ge |x|$ and $y \le -|x|+3$?
(Existing reasoning) Let's start by analyzing the given inequalities. The next
step is to determine the points of intersection... Now we know the points of
intersection are...

Output:
(Reward) 0.765
```

## 5 DEFINITION OF MDP

Combining the above components, our MDP is defined as follows:

- **State**: A low-dimensional vector generated from the LLM's self-evaluation of the current reasoning step.

- **Action**: One of five logic blocks that guide the next step reasoning.

- **Reward**: A score from a PRM that evaluates the quality of the reasoning step after an action is taken.

## 6 TRAINING PROCESS

Follow the MDP framework, we train the navigator with Deep Q-Learning algorithm. The training process is:

```
- For training episodes:
  - Randomly sample a Problem. Existing reasoning={}.
  - WHILE True:
    - Problem + Existing reasoning --[Self-evaluation]--> State vector
    - State vector --[MLP navigator]--> Logic block (action)
    - Problem + Existing reasoning --[Logic block]--> New reasoning
    - Existing reasoning <-- Existing reasoning + New reasoning
    - Problem + Existing reasoning --[PRM]--> Reward
    - Use the reward to train the MLP navigator
    - IF reach the answer: BREAK
```

## 7 INFERENCE PROCESS

With the trained navigator, the inference process is:

```
- Problem. Existing reasoning={}. Answers={}.
- For self-consistency candidates number:
  - WHILE True:
    - Problem + Existing reasoning --[Self-evaluation]--> State vector
    - State vector --[MLP navigator]--> Logic block (action)
    - Problem + Existing reasoning --[Logic block]--> New reasoning
    - Existing reasoning <-- Existing reasoning + New reasoning
    - IF reach the answer: BREAK
  - Answers <-- Answers + New answer
- Answers --[Self-consistency]--> Final answer
```

## 8 SELF-CONSISTENCY EVALUATION

To enhance the robustness of our final answers, we employ a self-consistency mechanism. Since the final answers can be in different formats, we use *sympy.parsing.latex* library to parse the answers and determine if they are mathematically equivalent. We select the final answer using majority voting. If no majority exists, one answer is selected at random.

Here is an input-output example for self-consistency:

```
Input:
(Answers) {0.5} {0.7} {1/2}

Output:
(Final answer) 0.5
```

# K PROMPTS

In this section, we provide the prompts for obtaining states and performing actions to ensure reproducibility. In the following prompt blocks, the text enclosed in braces "{}" denotes problem-specific content, such as intermediate reasoning steps, while the remaining texts serve as fixed templates.

## K.1 PROMPT FOR OBTAINING STATES

We use the following prompt to extract a state vector from an intermediate reasoning step. This prompt guides the LLM in systematically evaluating the current step across multiple aspects. During our experiments, we observed that LLMs effectively identify these aspects and provide detailed scores for each aspect.

Listing 1: Prompts for obtaining states

```
 1  {Problem and reasoning steps}
 2  Please evaluate the current step from the following aspects.
 3  A) Correctness
 4      A1: Correctness of modeling:
 5      Whether the current step is correctly derived from the origin
            problem.
 6      A2: Clarity for further reasoning:
 7      Whether the current step is clearly presented, without ambiguity
            , to support further reasoning.
 8      A3: Correctness of calculation:
 9      Whether the numerical computation in the current step is
            performed correctly.
10  B) Complexity
11      B1: Complexity to reach the final answer:
12      Whether it still requires complex reasoning or calculation to
            reach the final answer from the current step.
13      B2: Alternative methods in further reasoning:
14      Whether there exist multiple alternative methods to solve the
            problem in the current step.
15  C) Completeness
16      C1: Closeness to the final solution:
17      Whether the current step is close enough to directly reach the
            final answer.
18      C2: Completeness within the step:
19      Whether all necessary elements within this specific step are
            known from the problem or previous steps.
20  For each aspect, please score 1 for False, 2 for Unsure, and 3 for
        True, and score 0 if the current step does not involve this
        aspect. Please attach the reason for each score.
21  Use the format 'A1 score=[SCORE] reason=[REASON]'.
22  Only score the current reasoning step here, and DONOT conduct
        further reasoning.
```

## K.2 PROMPTS FOR ACTIONS

**Reason one step:** The prompt below is designed to conduct the action of "Reason one step". We emphasize that it should focus on one step at a time, which better controls the output and prevents mistakes in long reasoning paths.

Listing 2: Prompts for action "Reason one step"

```
 1  Here is a problem and several reasoning steps.
 2  {Problem and previous steps}
 3  Please reason exactly ONE more step based on the current step here,
        and DONOT reason too many steps at once.
```

**Decompose:** A "Decompose" action consists of the following three prompts. First, we use prompt 3 to break down the current problem into multiple subtasks. Next, prompt 4 is applied sequentially to

each subtask to complete its execution. Since the current subtask may depend on previous ones, the results of earlier subtasks are included (see line 4 of the prompt). Finally, the execution results are summarized using prompt 5, which captures the key steps and outcomes. Only the summarized result is utilized for subsequent reasoning actions.

Listing 3: Prompts for obtaining subtasks in action "Decompose"

```
1  Here is a problem and several reasoning steps.
2  {Problem and previous steps}
3  Please decompose the current task into subtasks, where we can solve
       the original problem by combining these results of subtasks.
4  Only provide subtasks decomposition here, and DONOT conduct specific
        reasoning or calculation.
5  Use the format '### Subtask1: subtask1'.
```

Listing 4: Prompts for executing subtasks in action "Decompose"

```
1  Here is a problem and several reasoning steps.
2  {Problem and reasoning steps before decomposition}
3  For the next step, the task is decomposed into subtasks, here are
       the reasonings in the first few subtasks.
4  {Executing results of previous subtasks}
5  Please conduct the following Subtask{subtask_id} to continue the
       reasoning.
6  DONOT conduct a more detailed decomposition for the subtask.
```

Listing 5: Prompts for summarize subtasks in action "Decompose"

```
1  Here are a few detailed reasoning subtasks of a problem.
2  {Executing results of subtasks}
3  Please give a clear and concise summary of these subtasks, keeping
       the key reasoning and results in each subtask.
4  Only provide the summary here, and DONOT conduct more reasoning or
       calculation.
```

**Debate:** A "Debate" action involves multiple rounds of question-answering. First, the LLM generates different plans for the task using prompt 6. Next, the plans are compared, and the most promising one is selected using prompt 7, mimicking how human experts debate and discuss to reach a solution. Based on the chosen plan, reasoning is advanced by one step through prompt 8. Similar to the "Decompose" action, only the results of the final one-step reasoning (output of prompt 8) are retained for subsequent reasoning processes.

Listing 6: Prompts for obtaining various plans in action "Debate"

```
1  Here is a problem and several reasoning steps.
2  {Problem and previous reasoning steps}
3  Please propose three different alternative plans for solving the
       problem in the current step.
4  Only provide plans here, and DONOT conduct specific reasoning or
       calculation.
5  Use the format '### Plan1: plan1'.
```

Listing 7: Prompts for analysing and comparing plans in action "Debate"

```
1  Here is a problem and several reasoning steps.
2  {Problem and previous reasoning steps}
3  Currently, we have several alternative plans for solving the problem
        in the current step.
4  {Generated Plans}
5  Please review and compare these plans carefully, and tell which one
       is most promising for further reasoning. Only compare the plans
       here, and DONOT conduct further reasoning or calculation.
6  Use the format 'The most promising plan is Plan[INDEX]: [REASON]',
       where [INDEX] is an integer index of the plan and [REASON] is a
       detailed analysis.
```

Listing 8: Prompts for executing the plan in action "Debate"

```
1  Here is a problem and several reasoning steps.
2  {Problem and previous reasoning steps}
3  For the next step, we have decided on the most promising plan:
4  {Plan}
5  Please reason **exactly one** more step according to the plan here,
     and DONOT reason too many steps at once.
```

**Refine:** We use prompt 9 to perform the "Refine" action, which instructs the LLM to review and improve the reasoning steps for clarity and correctness.

Listing 9: Prompts for action "Refine"

```
1  Here is a problem and several reasoning steps
2  {Problem and previous reasoning steps}
3  Please check and refine the current thought here, and DONOT conduct
     further reasoning or calculation.
```

**Terminate:** The prompt for the "Terminate" action concludes the reasoning process by generating the final answer. The only variation lies in the output format, which is adapted to the specific requirements of each dataset.

Listing 10: Prompts for action "Terminate"

```
1  Here is a problem and several reasoning steps
2  {Problem and previous reasoning steps}
3
4  ## GSM8K
5  Please generate the answer for the problem. Please end the answer
     with 'The answer is numerical_answer'.
6
7  ## MATH
8  Please generate the answer for the problem. Wrap the answer with \\
     boxed{{answer}}.
9
10 ## MMLU-STEM and GPQA
11 End the answer with 'The answer is (CHOICE)'.
12
13 ## StrategyQA
14 Please generate the answer for the problem. At the end of your
     answer, conclude the answer with 'The answer is yes' or 'The
     answer is no'.
```

### K.3 SENSITIVE TO ACTION-PROMPTS PHRASING.

To evaluate whether the performance of RLoT is sensitive to the specific wording of the action prompts, we conducted a robustness test by rephrasing the natural language instructions for the Decompose, Debate, and Refine actions.

**Decompose:** We rephrase the prompts for the action of Decompose as:

Listing 11: Prompts for obtaining subtasks in action "Decompose"

```
1  This is the problem and the reasoning progress so far.
2  {Problem and previous steps}
3  Break down the current problem into smaller, manageable sub-problems
     .
4  Solving these sub-problems sequentially should lead to the solution
     of the original question.
5  Only provide subtasks decomposition here, and DONOT conduct specific
      reasoning or calculation.
6  Format: '### Subtask1: subtask1'.
```

Listing 12: Prompts for executing subtasks in action "Decompose"

```
1  This is the problem context and the steps taken.
2  {Problem and reasoning steps before decomposition}
3  We have divided the main task into several parts. Below is the
       reasoning progress for the initial parts.
4  {Executing results of previous subtasks}
5  Now, please execute the specific instruction for Subtask{subtask_id}
        to advance the solution.
6  DONOT conduct a more detailed decomposition for the subtask.
```

Listing 13: Prompts for summarize subtasks in action "Decompose"

```
1  Below are the detailed execution steps for the problem's subtasks.
2  {Executing results of subtasks}
3  Synthesize the steps above into a brief and coherent overview,
       ensuring that the essential logic and final outcomes of each
       subtask are preserved.
4  Only provide the summary here, and DONOT conduct more reasoning or
       calculation.
```

**Debate:** We rephrase the prompts for the action of Debate as:

Listing 14: Prompts for obtaining various plans in action "Debate"

```
1  This is a problem and existing reasoning steps.
2  {Problem and previous reasoning steps}
3  Formulate three distinct strategies to address the problem from the
       current state.
4  Only provide plans here, and DONOT conduct specific reasoning or
       calculation.
5  Use the format '### Plan1: plan1'.
```

Listing 15: Prompts for analysing and comparing plans in action "Debate"

```
1  This is the problem context and the steps taken.
2  {Problem and previous reasoning steps}
3  We have generated a set of potential strategies to address the
       problem at this stage. {Generated Plans}
4  Critically evaluate these options and identify the most viable
       strategy to advance the reasoning.
5  Only compare the plans here, and DONOT conduct further reasoning or
       calculation.
6  Use the format 'The most promising plan is Plan[INDEX]: [REASON]',
       where [INDEX] is an integer index of the plan and [REASON] is a
       detailed analysis.
```

Listing 16: Prompts for executing the plan in action "Debate"

```
1  Here is a problem and several reasoning steps.
2  {Problem and previous reasoning steps}
3  We have selected the optimal strategy to proceed: {Plan}
4  Carry out the immediate next logical step following this strategy,
       and DONOT reason too many steps at once.
```

**Refine:** We rephrase the prompts for the action of Refine as:

Listing 17: Prompts for action "Refine"

```
1  Here is a problem and several reasoning steps
2  {Problem and previous reasoning steps}
3  Please examine the current reasoning for potential improvements and
       revise it accordingly, and DONOT conduct further reasoning or
       calculation.
```

Table 23: Impact of prompt rephrasing.

| LLM | Method | GSM8K | GPQA | StrategyQA | Average |
|---|---|---|---|---|---|
| Qwen2.5-Instruct-7B | DirectQA | 91.58 | 31.25 | 68.85 | 63.89 |
| | RLoT (original) | **92.87** | 44.64 | 79.04 | 72.18 |
| | RLoT (rephrased) | 92.49 | **45.09** | **79.91** | **72.42** |

The results, presented in Table 23, demonstrate that RLoT exhibits high robustness to prompt variations. The model with rephrased prompts achieves an average accuracy very similar to the original one. This consistency indicates that the effectiveness of RLoT stems from the adaptive logical structures constructed by the navigator, rather than overfitting to specific phrasing or reliance on extensive prompt engineering.

## L    USE OF LLMS

The authors used LLMs to aid or polish paper writing, but all content has been carefully reviewed by the author. The authors used LLMs for literature retrieval and discovery, but all related works have been carefully reviewed and organized by the author. The research ideation in this work was entirely completed by the author and does not involve the use of LLMs.

