# OpenReview forum: "RL of Thoughts: Navigating LLM Reasoning with Inference-time Reinforcement Learning"
_ICLR.cc/2026/Conference — ICLR 2026 Poster_

### Official Review · Reviewer_VvQn · 2025-10-28

**Soundness:** 3
**Presentation:** 1
**Contribution:** 2
**Rating:** 8
**Confidence:** 4

**Summary:**

The paper proposes the technique RL-of-Thoughts (RLoT), an inference-time method that trains a tiny “navigator” with reinforcement learning to compose task-adaptive reasoning structures for LLMs. Long-sequence reasoning is cast as an MDP, where the state is a self-evaluation of the current reasoning, and 5 possible actions: Reason one step, Decompose, Debate, Refine, and Terminate. A process-reward model scores intermediate steps to provide the reward signal. Experiments show consistent accuracy gains and the authors also claim transferability of a single navigator across LLMs and tasks.

**Strengths:**

1. It presents a method that reframes reasoning as an MDP over a small set of actions, avoiding deep tree searches.
2. Results show performance gains across different tasks and models. The authors have analyzed a wide range of models.
3. According to the authors, the method is transferable and generalizable to other models and datasets.

**Weaknesses:**

1. The method repeats inferences with self-consistency filtering. Including the results of token budgets per method and normalize across baselines can help understanding the budget costs in a fair way.
2.PRM calibration. How well do PRM scores (Math-Shepherd) correlate with eventual correctness on non-math tasks?
3. The paper fails to include confidence intervals of the results.

**Questions:**

1. How does the number of tokens generated to achieve the solution compare between RLoT and the baselines?
2. How does the number of actions affect the results? Why are these actions selected?

---

> ### Author Response · Authors · 2025-11-21
> **Rebuttal (Part 1/3)**
>
> We sincerely thank the reviewer for the positive evaluation and insightful suggestions.
> Below we provide a point-to-point response with additional experimental results and discussions.
>
> **Q1. Token budgets comparison**
>
> We thank the reviewer for this practical suggestion.
> To ensure a comparison of computational costs, we have reported the average token consumption per question in Appendix C, where the results of RLoT already incorporated the budget of self-consistency filtering.
> Also, per your suggestion, we normalize the overall average token budgets of different methods over Direct QA.
>
> | **Method** | **AIME24** |  | **AMC23** |  | **MATH** |  | **GSM8K** |  | **GPQA** |  | **MMLU-STEM** |  | **StrategyQA** |  | **Average** |  |
> |:---:|:---:|:---:|:---:|:---:|:---:|:---:|:---:|:---:|:---:|:---:|:---:|:---:|:---:|:---:|:---:|:---:|
> |  | Input | Output | Input | Output | Input | Output | Input | Output | Input | Output | Input | Output | Input | Output | Input | Output |
> | **Direct QA** | 1.00 | 1.00 | 1.00 | 1.00 | 1.00 | 1.00 | 1.00 | 1.00 | 1.00 | 1.00 | 1.00 | 1.00 | 1.00 | 1.00 | 1.00 | 1.00 |
> | **Zero-shot CoT** | 1.03 | 1.01 | 1.04 | 1.01 | 1.10 | 0.98 | 1.09 | 1.07 | 1.03 | 1.00 | 1.12 | 1.05 | 1.07 | 1.32 | 1.05 | 1.02 |
> | **Few-shot CoT** | 3.75 | 0.75 | 3.99 | 1.04 | 5.43 | 0.78 | 10.10 | 0.94 | 7.27 | 0.98 | 7.94 | 0.86 | 5.51 | 1.07 | 5.26 | 0.87 |
> | **CoT-SC** | 14.90 | 3.07 | 15.99 | 4.37 | 21.91 | 2.95 | 40.34 | 3.78 | 29.06 | 3.83 | 30.98 | 3.51 | 21.69 | 4.28 | 20.95 | 3.53 |
> | **ToT** | 7.85 | 3.66 | 7.31 | 7.78 | 31.59 | 7.61 | 35.95 | 21.71 | 18.50 | 5.57 | 27.02 | 9.41 | 50.71 | 12.01 | 16.22 | 6.69 |
> | **RLoT (ours)** | 30.69 | 2.13 | 27.15 | 5.00 | 23.81 | 2.13 | 23.64 | 7.31 | 17.43 | 3.40 | 20.58 | 3.66 | 29.50 | 6.19 | 26.13 | 3.36 |
>
> The results demonstrate that RLoT improves the reasoning ability of LLMs with a low cost comparing with complex inference-time techniques.
> This is because the trained navigator can directly select appropriate logic blocks without searching like ToT, improving the performance and reducing the output cost.
>
>
> **Q2. PRM calibration**
>
> We thank the reviewer for raising this critical point.
> We agree that since we utilize a math-trained PRM (Math-Shepherd) for non-math benchmarks like GPQA (STEM) and StrategyQA (Commonsense), validating its generalization capability is essential.
>
> To empirically verify this, we conducted a PRM calibration analysis detailed in Appendix H.2.
> We calculated the Pearson's correlation coefficient between the intermediate PRM scores assigned at various reasoning steps and the correctness of the final answer.
>
> | **LLM** | **Correlation with Correctness** | **MATH** | **GPQA** | **StrategyQA** |
> |:---:|:---:|:---:|:---:|:---:|
> | Qwen2.5-Instruct-7B | Step1 PRM | 0.199 | 0.184 | 0.157 |
> |  | Step2 PRM | 0.222 | 0.213 | 0.195 |
> |  | Step3 PRM | 0.236 | 0.215 | 0.202 |
> |  | Step4 PRM | 0.240 | 0.228 | 0.218 |
> |  | Average | 0.224 | 0.210 | 0.193 |
>
> The results indicate a consistent positive correlation across all different benchmarks.
> This demonstrates that the Math-Shepherd PRM captures not just mathematical specificities, but also the intrinsic logic of step-by-step reasoning in other related tasks.
> This also echoes recent study suggesting that process reward models (PRMs) trained on mathematical data exhibit strong transferability to related domains such as STEM [1].
>
> [1] Zhang et al. The lessons of developing process reward models in mathematical reasoning. ACL (Findings) 2025.

---

> ### Author Response · Authors · 2025-11-21
> **Rebuttal (Part 2/3)**
>
> **Q3. Confidence intervals**
>
> We acknowledge that reporting variance is crucial for fair comparison, although we initially followed the convention of recent technical reports (e.g., Qwen, Gemini, DeepSeek) which often omit variance due to space constraints.
>
> To address your concern, we have conducted 5 repeated runs and reported the mean ± standard deviation in the Qwen LLMs in our main results (Table 2):
>
> | **LLM** | **Method** | **Olympic math** |  | **Elementary math** |  | **STEM** |  | **Commonsense** | **Average** |
> |:---:|:---:|:---:|:---:|:---:|:---:|:---:|:---:|:---:|:---:|
> |  |  | AIME24 | AMC23 | MATH | GSM8K | GPQA | MMLU-STEM | StrategyQA |  |
> | Qwen2.5-14B-Instruct | Direct QA | 13.33±0.00 | 56.50±1.22 | 78.55±0.40 | 93.95±1.04 | 36.74±0.63 | 85.01±0.23 | 72.90±0.80 | 62.43±0.18 |
> |  | Zero-shot CoT | 16.67±2.98 | 62.50±1.58 | 78.59±0.39 | 94.22±0.65 | 37.99±0.55 | 85.54±0.62 | 76.30±0.78 | 64.55±0.70 |
> |  | Few-shot CoT | 6.67±0.00 | 55.50±1.87 | 79.91±0.41 | **94.39±0.91** | 45.18±0.85 | 85.18±0.47 | 78.05±0.46 | 63.55±0.19 |
> |  | CoT-SC | 6.67±0.00 | 45.50±1.00 | 80.08±0.47 | 93.66±0.35 | 45.13±0.74 | 86.48±0.45 | 79.91±1.03 | 62.49±0.25 |
> |  | ToT | 10.00±0.00 | 55.50±1.87 | 79.71±0.43 | 93.83±1.44 | 45.36±0.71 | 86.51±0.37 | 78.49±0.87 | 64.20±0.31 |
> |  | **RLoT (ours)** | **24.67±1.63***** | **65.50±1.87***** | **80.47±0.39***** | 94.24±1.01 | **51.43±0.94***** | **89.10±0.45***** | **80.47±0.62***** | **69.41±0.35***** |
> | Qwen2.5-7B-Instruct | Direct QA | 10.00±0.00 | 41.00±1.22 | 74.78±0.43 | 91.54±1.07 | 31.47±0.40 | 80.85±0.14 | 68.94±0.76 | 56.94±0.23 |
> |  | Zero-shot CoT | 16.00±2.49 | 54.00±1.22 | 74.88±0.23 | 91.81±1.03 | 34.33±0.62 | 80.93±0.45 | 70.92±0.47 | 60.41±0.49 |
> |  | Few-shot CoT | 13.33±0.00 | 46.00±1.22 | 75.28±0.28 | 91.73±0.66 | 36.47±0.61 | 81.39±0.40 | 73.86±0.43 | 59.72±0.22 |
> |  | CoT-SC | 6.67±0.00 | 51.00±1.22 | 76.12±0.35 | 92.01±1.03 | 38.62±0.49 | 83.20±0.33 | 77.85±0.62 | 60.78±0.15 |
> |  | ToT | 13.33±0.00 | 51.00±1.22 | 73.74±0.51 | 91.83±0.80 | 36.43±0.57 | 82.61±0.40 | 70.86±0.96 | 59.97±0.24 |
> |  | **RLoT (ours)** | **22.67±1.33***** | **60.00±1.58***** | **76.84±0.21***** | **92.95±0.63***** | **44.78±0.71***** | **84.90±0.35***** | **79.56±0.64***** | **65.96±0.29***** |
>
> and the extended baseline comparisons the reviewer pointed out (original Table 9, now Table 10):
>
> | **LLM** | **Method** |  | **MATH-500** | **GSM8K** | **GPQA** | **StrategyQA** | **Average** |
> |:---:|:---:|:---:|:---:|:---:|:---:|:---:|:---:|
> | Qwen2.5-Instruct-7B | DirectQA |  | 74.96±1.37 | 90.95±1.06 | 31.16±0.23 | 68.53±0.53 | 66.40±0.46 |
> |  | Self-Refine |  | 72.32±1.20 | 88.39±0.38 | 32.23±0.61 | 74.06±0.79 | 66.75±0.33 |
> |  | Least-to-Most |  | 73.28±0.84 | 89.10±0.82 | 31.79±0.54 | 73.60±0.64 | 66.94±0.25 |
> |  | SelfCheck |  | 74.36±1.60 | 90.24±1.09 | 33.84±0.61 | 72.87±0.77 | 67.83±0.35 |
> |  | DeAR |  | 69.92±0.88 | 87.84±0.54 | 35.71±0.37 | 71.03±0.85 | 66.13±0.39 |
> |  | Tree search | One-step-greedy | 71.24±1.43 | 89.49±0.99 | 32.63±0.50 | 72.43±0.56 | 66.45±0.54 |
> |  |  | Basic MCTS | 72.40±0.79 | 89.51±0.72 | 35.13±0.59 | 75.72±0.80 | 68.19±0.19 |
> |  |  | Litesearch | 72.52±1.29 | 89.86±0.86 | 33.17±0.30 | 76.04±0.65 | 67.90±0.52 |
> |  |  | Q* | 73.76±0.81 | 92.28±0.61 | 34.38±0.76 | 76.80±0.99 | 69.30±0.50 |
> |  |  | rStar | 75.88±1.02 | 92.05±0.89 | 38.17±0.32 | 77.09±0.83 | 70.80±0.68 |
> |  |  | AFlow | 75.72±1.16 | 90.95±0.98 | 37.37±0.63 | 77.09±0.56 | 70.28±0.13 |
> |  | DSPy |  | 72.48±0.86 | 88.75±0.77 | 34.11±0.68 | 73.89±0.94 | 67.31±0.49 |
> |  | Graph of Thoughts (GoT) |  | 73.24±1.06 | 88.76±1.07 | 33.26±0.66 | 75.69±0.91 | 67.74±0.39 |
> |  | Buffer of Thoughts (BoT) |  | 75.16±0.74 | 91.98±0.64 | 34.91±0.58 | 74.38±0.44 | 69.11±0.32 |
> |  | **RLoT (ours)** |  | **76.88±1.35***** | **92.95±0.63***** | **44.78±0.71***** | **79.56±0.64***** | **73.54±0.18***** |
>
> We are working on repeated tests for other results.
> Our repeated experiments reveal that the variance in total scores is actually quite small.
> This stability can be attributed to the large size of the test sets (e.g., 5000 for MATH) and the relatively consistent performance of the model on specific problem types.
> We performed statistical significance tests (two-sided t-tests), and the results indicate that the performance gaps between RLoT and the best-performing baselines are statistically significant (***p < 0.01) across most datasets.

---

> ### Author Response · Authors · 2025-11-21
> **Rebuttal (Part 3/3)**
>
> **Q4. Design and effect of actions**
>
> We thank the reviewer for this question.
>
> **First**, we have investigated the relationship between the maximum number of allowed logic blocks (actions) and the final performance in Appendix I.1.
> As illustrated in Figure 8, we evaluated the performance on GSM8K, GPQA, and StrategyQA by varying the maximum number of logic blocks from 1 to 5.
> The results demonstrate a positive correlation: a longer sequence of logic blocks leads to performance gains across all tested benchmarks.
>
> **Second**, the blocks were designed by cognitive theory on human reasoning [1].
> Also, they are proved to be effective in previous works [2,3].
> "Decompose," "Debate," and "Refine" are the core actions, supplemented by "Reason one step" and "Terminate" to ensure complete reasoning flows.
>
> To validate the effectiveness of each blocks, we remove each of the core blocks and train a navigator (Section 4.7):
>
> |LLM|Ablation|MATH|GPQA|StrategyQA|Average|
> |:-:|:-:|:-:|:-:|:-:|:-:|
> |Qwen2.5-Instruct-7B|w/o Decompose|75.42|31.92|77.00|61.45|
> ||w/o Debate|74.02|36.61|77.58|62.74|
> ||w/o Refine|75.76|41.29|72.93|63.33|
> ||**Full RLoT**|**76.70**|**44.64**|**79.04**|**66.79**|
> |GPT-4o-mini|w/o Decompose|76.08|41.74|79.33|65.72|
> ||w/o Debate|74.68|45.31|78.75|66.25|
> ||w/o Refine|76.44|51.56|74.38|67.46|
> ||**Full RLoT**|**77.36**|**54.02**|**82.68**|**71.35**|
>
> The ablation results confirm that all blocks are effective.
>
> **Third**, we noticed that different tasks rely more on different blocks.
> We calculate the usage frequency (%) of each logic block across different tasks, averaged over all tested models. The results are presented below:
>
> |Logic block|AIME24|AMC23|MATH|GSM8K|GPQA|MMLU-STEM|StrategyQA|Total|
> |:-:|:-:|:-:|:-:|:-:|:-:|:-:|:-:|:-:|
> |Reason-one-step|47.98|47.19|47.27|43.48|58.68|56.31|74.03|53.56|
> |Decompose|0.58|0.00|0.41|0.26|0.51|1.47|1.17|0.63|
> |Debate|0.58|0.87|0.32|0.01|2.49|6.67|4.61|2.22|
> |Refine|33.53|34.63|33.47|37.97|17.69|12.52|2.68|24.64|
> |Terminate|17.34|17.32|18.53|18.27|20.64|23.03|17.51|18.95|
>
> From the results, we observe that the usage frequency of different blocks is not uniform.
> As a fundamental action, "Reason one step" is used most often.
> Among the three core actions, "Refine" is the most frequently selected one.
> Besides, the navigator adapts its strategy based on the task type.
> For instance, "Refine" is used heavily in mathematical problems, which often require careful checking of calculations.
> In contrast, for STEM and commonsense reasoning tasks, the use of "Refine" decreases while "Debate" is employed more often.
>
> [1] On the cognitive process of human problem solving. Cognitive systems research 2010.\
> [2] Self-Refine: Iterative refinement with self-feedback. NeurIPS 2023.\
> [3] Encouraging divergent thinking in large language models through multi-agent debate. arxiv 2023.
>
> \
> \
> Thanks again for your valuable suggestions.
> We have made brief revisions to the manuscript accordingly and will continue to work to incorporate the above discussions and improvements into the final version.

---

> ### Comment · Reviewer_VvQn · 2025-11-27
>
> Thank you for addressing my concerns and including the corresponding content. The results variance should still be included in the paper. The effect of different blocks is also a good addition to the analysis and you should consider including it in the paper.
>
> I believe this paper makes a good contribution to the methods of RL in LLMs. My score already reflects my thoughts.

---

> > ### Author Response · Authors · 2025-11-27
> >
> > We will include the discussions and results variance in our final version. Thanks again for your response and suggestions!

---

### Official Review · Reviewer_HceF · 2025-10-30

**Soundness:** 3
**Presentation:** 3
**Contribution:** 3
**Rating:** 6
**Confidence:** 4

**Summary:**

The paper proposes RLoT, a new inference-time technique to enhance the complex reasoning capabilities of LLMs. The core problem addressed is that existing inference-time methods use predefined, task-agnostic logical structures that lack adaptability. RLoT's key idea is to train a lightweight "navigator model" using Reinforcement Learning to dynamically construct task-adaptive logical structures during inference.

**Strengths:**

1. The primary strength is the RLoT framework itself. It reframes the problem from "how to build a better reasoning LLM" or "what is the best fixed reasoning structure" to "how to learn a policy that navigates a frozen LLM's reasoning space." This is a clever and effective conceptual leap.

2. The results show that a sub-10B model with a 3K-parameter navigator can close most of the performance gap to a 70B model on tasks like GPQA and StrategyQA is a massive win.

3. The method is tested on multiple SOTA LLMs and a wide array of difficult reasoning benchmarks.

**Weaknesses:**

1. The state space, which is the sole input to the RL agent, is based on the LLM's own self-evaluation. The authors provide a validation (82% accuracy) in Appendix F and show it works better than alternatives. However, this remains a potential point of noise or failure. One might question if an LLM that is failing at a reasoning task (the "hard questions" used for training ) can simultaneously be a reliable evaluator of its own failing state. This could be particularly problematic for smaller, less-capable models.

2. The five logic blocks are "inspired from human cognition"  and the quality of the entire system is upper-bounded by the quality and expressiveness of this predefined action set. However, they are hand-engineered component.

**Questions:**

1. The quality of the 5 logic blocks seems highly dependent on the quality of their prompts (listed in Appendix K.2 ). How much prompt-engineering was required to make these actions (especially Debate and Refine ) work reliably? Is the framework sensitive to the phrasing of these action-prompts?

---

> ### Author Response · Authors · 2025-11-21
> **Rebuttal**
>
> We sincerely thank the reviewer for the positive evaluation and insightful suggestions.
> Below we provide a point-to-point response with additional experimental results and discussions.
>
> **Q1. Reliability of self-evaluation state**
>
> Thanks for this important point.
>
> To explore the impact of noisy states, we conducted a quantitative analysis in Appendix F.2, where we deliberately introduced synthetic noise into the self-evaluation state vectors during inference.
>
> | **LLM** | **State** | **GSM8K** | **GPQA** | **StrategyQA** | **Average** |
> |:---:|:---:|:---:|:---:|:---:|:---:|
> | Qwen2.5-Instruct-7B | DirectQA | 91.58 | 31.25 | 68.85 | 63.89 |
> |  | 50% noise | 91.74 | 34.82 | 68.41 | 64.99 |
> |  | 30% noise | 92.27 | 43.75 | 71.32 | 69.11 |
> |  | **Normal** | **92.87** | **44.64** | **79.04** | **72.18** |
>
> The framework exhibits robustness to noise.
> Even when 50% of the state bits were randomly flipped, RLoT maintained an average accuracy of 64.99%, which is still comparable to the Direct QA baseline.
> As the noise decreased to 30%, the performance recovered rapidly to 69.11%, showing clear improvements over the baseline.
>
> Meanwhile, to minimize potential noise inherent in self-evaluation, we deliberately designed the evaluation criteria as a binary classification task rather than a multi-grade continuous scoring.
> Continuous or fine-grained scoring (e.g., 1-10 scales) requires precise calibration, which is challenging for LLMs.
> In contrast, a binary standard simplifies the decision boundary, significantly reducing the cognitive load on the LLM.
> This design choice makes the evaluation process a easier task than reasoning itself, ensuring that the state vector remains a stable and reliable signal for the navigator even with less capable smaller LLMs.
>
> We performed empirical statistics to illustrate this.
> We re-evaluated the intermediate steps obtained by Qwen2.5-Instruct-7B on three datasets using Qwen2.5-Instruct-3B with the same prompt.
> Below is the overlap between the self-evaluation of Qwen2.5-Instruct-7B and the external evaluation of Qwen2.5-Instruct-3B.
>
> | Aspect |  | **MATH** | **GPQA** | **StrategyQA** | **Average** |
> |:---:|:---:|:---:|:---:|:---:|:---:|
> | **A:Correctness** | A1 | 91.26 | 88.17 | 95.23 | 91.55 |
> |  | A2 | 93.76 | 89.12 | 95.68 | 92.85 |
> |  | A3 | 88.19 | 85.91 | 92.70 | 88.93 |
> | **B:Complexity** | B1 | 86.91 | 83.01 | 89.69 | 86.54 |
> |  | B2 | 87.04 | 82.15 | 89.45 | 86.21 |
> | **C:Completeness** | C1 | 85.94 | 83.24 | 90.17 | 86.45 |
> |  | C2 | 90.49 | 87.62 | 94.05 | 90.72 |
> | **Average** |  | 89.08 | 85.60 | 92.42 | 89.04 |
>
> The relatively high degree of overlap indicates that smaller LLMs can also achieve usable results under our design of binary evaluation standards.
>
>
> **Q2. Sensitivity to prompt-engineering**
>
> We thank the reviewer for this important point.
>
> In our actual research process, we did not extensively tune or iterate on the prompts.
> Instead, we just designed a set of straightforward prompts based on standard human cognitive descriptions and used them directly for all experiments.
>
> To empirically verify, we conducted a sensitivity test.
> We invited an independent colleague to rephrase the natural language instructions for the three most complex actions: Decompose, Debate, and Refine, as shown in Appendix K.3.
> We then tested the RLoT framework using these rephrased prompts.
>
> | **LLM** | **Method** | **GSM8K** | **GPQA** | **StrategyQA** | **Average** |
> |:---:|:---:|:---:|:---:|:---:|:---:|
> | Qwen2.5-Instruct-7B | DirectQA | 91.58 | 31.25 | 68.85 | 63.89 |
> |  | RLoT (original) | **92.87** | 44.64 | 79.04 | 72.18 |
> |  | RLoT (rephrased) | 92.49 | **45.09** | **79.91** | **72.42** |
>
> The results show that the performance remained very similar:
> This demonstrates that the effectiveness of RLoT stems from the adaptive logical structures generated by the RL Navigator, rather than overfitting to specific prompt phrasing or reliance on extensive prompt engineering.
>
> \
> \
> Thanks again for your valuable suggestions.
> We have made brief revisions to the manuscript accordingly and will continue to work to incorporate the above discussions and improvements into the final version.

---

### Official Review · Reviewer_TZiL · 2025-11-01

**Soundness:** 3
**Presentation:** 3
**Contribution:** 3
**Rating:** 6
**Confidence:** 3

**Summary:**

The paper proposes RL-of-Thoughts (RLoT): treat multi-step reasoning as an MDP where a tiny navigator (Double-Dueling DQN, <3k params) learns to select among five logic blocks (Reason one step, Decompose, Debate, Refine, Terminate) while the base LLM stays frozen. States are 7-aspect self-evaluations the LLM generates via a fixed prompt; rewards come from a Process Reward Model (PRM), e.g. math shepherd. The navigator is trained on "hard" questions only; at test time they use self-consistency to pick the final answer.

**Strengths:**

- Reasoning as an MDP with a standardized action set and state interface; prompts for states/actions are fully specified and the MDP is clearly defined. The navigator adds negligible runtime overhead; training cost is amortized and reported with token accounting.
- Good comparison with baselines (fixed workflows like DeAR, search methods like GoT, r\*, LiteSearch, DSPy, ...) with an "extended table" across GSM8K/GPQA/StrategyQA; The authors also report token usage vs. accuracy.
- Proper ablation:  Remove-one-block studies (for Decompose, Debate, and Refine),

**Weaknesses:**

Major problem is the insignificance of many results: For instance in Table 9, RLoT = 92.87 while several baselines cluster ≥92% (e.g. Buffer-of-Thoughts 92.35,  ...). The paper text claims overall wins, but GSM8K specifically looks like parity rather than a clear lead; without error bars the <1% differences are hard to interpret and they are rather frequent in the paper.

**Questions:**

1. Could you report mean ± 95% CI over ≥3 seeds for the results (especially Table 9) (GSM8K/GPQA/StrategyQA) and key math sets? If not, please justify why single-seed is sufficient given self-consistency randomness.
2. Is there a reason behind fix training at 3,000 episodes? Do curves show continued improvement (or overfitting) at 1k/5k?
3. Table 10 reports tokens/question, but wall-clock or FLOPs can be very informative as different methods assume different number of model. Can you add runtime and PRM calls to enable compute-fair comparisons vs. search-heavy methods (e.g., r\*, LiteSearch)?
4. Did you think of including SelfCheck[1] and Least-to-Most prompting [2] to experiments as baselines? Even a lightweight SelfCheck (PRM as a one-shot checker without RL) would clarify how much the RL policy contributes beyond access to a PRM.
5. You ablate −Decompose/−Debate/−Refine; can you also show fixed schedules (e.g., Decompose→Reason→Refine cycles à la DeAR) under the same token budget to separate "RL policy" gains from "having blocks at all"?

[1] Miao et. al.. Selfcheck: Using llms to zero-shot check their own step-by-step reasoning. ICLR, 2023.

[2] Zhou,et al. Least-to-most prompting enables complex reasoning in large language models. ICLR, 2022.

---

> ### Author Response · Authors · 2025-11-21
> **Rebuttal (Part 1/3)**
>
> We sincerely thank the reviewer for the positive evaluation and insightful suggestions.
> Below we provide a point-to-point response with additional experimental results and discussions.
>
> **Q1. Confidence intervals and results significance**
>
> We thank the reviewer for the rigorous suggestion regarding statistical significance.
> We acknowledge that reporting variance is crucial for fair comparison, although we initially followed the convention of recent technical reports (e.g., Qwen, Gemini, DeepSeek) which often omit variance due to space constraints.
>
> To address your concern, we have conducted 5 repeated runs and reported the mean ± standard deviation in the Qwen LLMs in our main results (Table 2):
>
> | **LLM** | **Method** | **Olympic math** |  | **Elementary math** |  | **STEM** |  | **Commonsense** | **Average** |
> |:---:|:---:|:---:|:---:|:---:|:---:|:---:|:---:|:---:|:---:|
> |  |  | AIME24 | AMC23 | MATH | GSM8K | GPQA | MMLU-STEM | StrategyQA |  |
> | Qwen2.5-14B-Instruct | Direct QA | 13.33±0.00 | 56.50±1.22 | 78.55±0.40 | 93.95±1.04 | 36.74±0.63 | 85.01±0.23 | 72.90±0.80 | 62.43±0.18 |
> |  | Zero-shot CoT | 16.67±2.98 | 62.50±1.58 | 78.59±0.39 | 94.22±0.65 | 37.99±0.55 | 85.54±0.62 | 76.30±0.78 | 64.55±0.70 |
> |  | Few-shot CoT | 6.67±0.00 | 55.50±1.87 | 79.91±0.41 | **94.39±0.91** | 45.18±0.85 | 85.18±0.47 | 78.05±0.46 | 63.55±0.19 |
> |  | CoT-SC | 6.67±0.00 | 45.50±1.00 | 80.08±0.47 | 93.66±0.35 | 45.13±0.74 | 86.48±0.45 | 79.91±1.03 | 62.49±0.25 |
> |  | ToT | 10.00±0.00 | 55.50±1.87 | 79.71±0.43 | 93.83±1.44 | 45.36±0.71 | 86.51±0.37 | 78.49±0.87 | 64.20±0.31 |
> |  | **RLoT (ours)** | **24.67±1.63***** | **65.50±1.87***** | **80.47±0.39***** | 94.24±1.01 | **51.43±0.94***** | **89.10±0.45***** | **80.47±0.62***** | **69.41±0.35***** |
> | Qwen2.5-7B-Instruct | Direct QA | 10.00±0.00 | 41.00±1.22 | 74.78±0.43 | 91.54±1.07 | 31.47±0.40 | 80.85±0.14 | 68.94±0.76 | 56.94±0.23 |
> |  | Zero-shot CoT | 16.00±2.49 | 54.00±1.22 | 74.88±0.23 | 91.81±1.03 | 34.33±0.62 | 80.93±0.45 | 70.92±0.47 | 60.41±0.49 |
> |  | Few-shot CoT | 13.33±0.00 | 46.00±1.22 | 75.28±0.28 | 91.73±0.66 | 36.47±0.61 | 81.39±0.40 | 73.86±0.43 | 59.72±0.22 |
> |  | CoT-SC | 6.67±0.00 | 51.00±1.22 | 76.12±0.35 | 92.01±1.03 | 38.62±0.49 | 83.20±0.33 | 77.85±0.62 | 60.78±0.15 |
> |  | ToT | 13.33±0.00 | 51.00±1.22 | 73.74±0.51 | 91.83±0.80 | 36.43±0.57 | 82.61±0.40 | 70.86±0.96 | 59.97±0.24 |
> |  | **RLoT (ours)** | **22.67±1.33***** | **60.00±1.58***** | **76.84±0.21***** | **92.95±0.63***** | **44.78±0.71***** | **84.90±0.35***** | **79.56±0.64***** | **65.96±0.29***** |
>
> and the extended baseline comparisons the reviewer pointed out (original Table 9, now Table 10):
>
> | **LLM** | **Method** |  | **MATH-500** | **GSM8K** | **GPQA** | **StrategyQA** | **Average** |
> |:---:|:---:|:---:|:---:|:---:|:---:|:---:|:---:|
> | Qwen2.5-Instruct-7B | DirectQA |  | 74.96±1.37 | 90.95±1.06 | 31.16±0.23 | 68.53±0.53 | 66.40±0.46 |
> |  | Self-Refine |  | 72.32±1.20 | 88.39±0.38 | 32.23±0.61 | 74.06±0.79 | 66.75±0.33 |
> |  | Least-to-Most |  | 73.28±0.84 | 89.10±0.82 | 31.79±0.54 | 73.60±0.64 | 66.94±0.25 |
> |  | SelfCheck |  | 74.36±1.60 | 90.24±1.09 | 33.84±0.61 | 72.87±0.77 | 67.83±0.35 |
> |  | DeAR |  | 69.92±0.88 | 87.84±0.54 | 35.71±0.37 | 71.03±0.85 | 66.13±0.39 |
> |  | Tree search | One-step-greedy | 71.24±1.43 | 89.49±0.99 | 32.63±0.50 | 72.43±0.56 | 66.45±0.54 |
> |  |  | Basic MCTS | 72.40±0.79 | 89.51±0.72 | 35.13±0.59 | 75.72±0.80 | 68.19±0.19 |
> |  |  | Litesearch | 72.52±1.29 | 89.86±0.86 | 33.17±0.30 | 76.04±0.65 | 67.90±0.52 |
> |  |  | Q* | 73.76±0.81 | 92.28±0.61 | 34.38±0.76 | 76.80±0.99 | 69.30±0.50 |
> |  |  | rStar | 75.88±1.02 | 92.05±0.89 | 38.17±0.32 | 77.09±0.83 | 70.80±0.68 |
> |  |  | AFlow | 75.72±1.16 | 90.95±0.98 | 37.37±0.63 | 77.09±0.56 | 70.28±0.13 |
> |  | DSPy |  | 72.48±0.86 | 88.75±0.77 | 34.11±0.68 | 73.89±0.94 | 67.31±0.49 |
> |  | Graph of Thoughts (GoT) |  | 73.24±1.06 | 88.76±1.07 | 33.26±0.66 | 75.69±0.91 | 67.74±0.39 |
> |  | Buffer of Thoughts (BoT) |  | 75.16±0.74 | 91.98±0.64 | 34.91±0.58 | 74.38±0.44 | 69.11±0.32 |
> |  | **RLoT (ours)** |  | **76.88±1.35***** | **92.95±0.63***** | **44.78±0.71***** | **79.56±0.64***** | **73.54±0.18***** |
>
> We are working on repeated tests for other results.
> Our repeated experiments reveal that the variance in total scores is actually quite small.
> This stability can be attributed to the large size of the test sets (e.g., 5000 for MATH) and the relatively consistent performance of the model on specific problem types.
> We performed statistical significance tests (two-sided t-tests), and the results indicate that the performance gaps between RLoT and the best-performing baselines are statistically significant (***p < 0.01) across most datasets.

---

> ### Author Response · Authors · 2025-11-21
> **Rebuttal (Part 2/3)**
>
> **(Q1 continued)**
>
> Your observation is true that GSM8K results appear close to parity (e.g., RLoT 92.87% vs. baselines ~92%).
> This may because GSM8K is relatively easy, making it difficult to demonstrate a clear lead.
> In contrast, the advantages of RLoT are much more pronounced on challenging benchmarks where baselines struggle, such as GPQA and AIME24.
> We have also revised the Abstract and Introduction to mitigate our claims, ensuring they accurately reflect that while RLoT is competitive on simpler tasks, its primary strength lies in navigating complex reasoning spaces for harder problems.
>
>
> **Q2. Training episodes and convergency**
>
> Thanks for this question.
>
> To justify the emperical setting of 3,000 episodes training, we have added experimental analyses regarding the RL training process in Appendix A.2.
> We investigate policy stability by comparing Double-Dueling DQN (DDDQN) with a standard Double DQN (DDQN), and we extended the training duration to 5,000 episodes.
>
> | **LLM** | **Method** | **Olympic math** |  | **Elementary math** |  | **STEM** |  | **Commonsense** | **Average** |
> |:---:|:---:|:---:|:---:|:---:|:---:|:---:|:---:|:---:|:---:|
> |  |  | AIME24 | AMC23 | MATH | GSM8K | GPQA | MMLU-STEM | StrategyQA |  |
> | Qwen2.5-14B-Instruct | Direct QA | 13.33±0.00 | 56.50±1.22 | 78.55±0.40 | 93.95±1.04 | 36.74±0.63 | 85.01±0.23 | 72.90±0.80 | 62.43±0.18 |
> |  | Zero-shot CoT | 16.67±2.98 | 62.50±1.58 | 78.59±0.39 | 94.22±0.65 | 37.99±0.55 | 85.54±0.62 | 76.30±0.78 | 64.55±0.70 |
> |  | Few-shot CoT | 6.67±0.00 | 55.50±1.87 | 79.91±0.41 | 94.39±0.91 | 45.18±0.85 | 85.18±0.47 | 78.05±0.46 | 63.55±0.19 |
> |  | CoT-SC | 6.67±0.00 | 45.50±1.00 | 80.08±0.47 | 93.66±0.35 | 45.13±0.74 | 86.48±0.45 | 79.91±1.03 | 62.49±0.25 |
> |  | ToT | 10.00±0.00 | 55.50±1.87 | 79.71±0.43 | 93.83±1.44 | 45.36±0.71 | 86.51±0.37 | 78.49±0.87 | 64.20±0.31 |
> |  | RLoT (DDDQN-3k) | **24.67±1.63** | **65.50±1.87** | **80.47±0.39** | 94.24±1.01 | 51.43±0.94 | 89.10±0.45 | 80.47±0.62 | **69.41±0.35** |
> |  | RLoT (DDDQN-5k) | 22.67±2.49 | 64.00±1.22 | 80.32±0.44 | **94.33±0.52** | **52.14±0.67** | 89.01±0.42 | **80.73±1.17** | 69.03±0.42 |
>
> As shown in Figure 4 and the above table, while the Double DQN eventually converges to a comparable performance level, it exhibits slightly decreased training stability compared to the Double-Dueling architecture.
> This empirical evidence supports our decision to use Double-Dueling DQN as the default algorithm.
> The results also show that the curves keep rising at 1,000 episodes, but essentially reache convergence around 3,000 episodes with only marginal gains thereafter, confirming our setting of training episodes.
>
>
> **Q3. Runtime comparison with baselines**
>
> We agree with the reviewer that token consumption alone does not reflect the total computational cost, especially given the structural differences between methods.
> For instance, search-based methods like LiteSearch incur additional overhead from frequent PRM calls during inference, whereas RLoT does not require any PRM calls at inference, it incurs a minor overhead from the Navigator's forward pass.
>
> To enable a compute-fair comparison, we have added the solving time (minutes per question), which captures the end-to-end latency, including LLM token generation, Navigator inference, and PRM calls (if applicable).
>
> | **LLM** | **Method** | **GSM8K** |  |  | **GPQA** |  |  | **StrategyQA** |  |  | **Average** |  |  |
> |:---:|:---:|:---:|:---:|:---:|:---:|:---:|:---:|:---:|:---:|:---:|:---:|:---:|:---:|
> |  |  | Score | Token | Time | Score | Token | Time | Score | Token | Time | Score | Token | Time |
> | Qwen2.5-Instruct-7B | DirectQA | 91.58 | 240 | 0.05 | 31.25 | 705 | 0.17 | 68.85 | 301 | 0.05 | 63.89 | 415 | 0.09 |
> |  | Self-Refine | 88.55 | 678 | 0.12 | 32.59 | 1784 | 0.41 | 73.94 | 939 | 0.16 | 65.03 | 1134 | 0.23 |
> |  | Litesearch | 89.76 | 912 | 0.20 | 33.04 | 2139 | 0.55 | 75.84 | 1228 | 0.24 | 66.21 | 1426 | 0.33 |
> |  | rStar | 91.96 | 2636 | 0.77 | 38.62 | 4127 | 1.44 | 77.15 | 2987 | 0.82 | 69.24 | 3483 | 1.01 |
> |  | RLoT (ours) | 92.87 | 1759 | 0.61 | 44.64 | 3001 | 0.95 | 79.04 | 2341 | 0.53 | 72.18 | 2367 | 0.70 |
>
> The results demonstrate that RLoT achieves superior performance with a similar or lower time cost compared to search-heavy baselines.
> Unlike methods that rely on extensive trial-and-error or tree search, our Navigator directly generates task-specific logical trajectories, effectively balancing high reasoning performance with inference efficiency.

---

> ### Author Response · Authors · 2025-11-21
> **Rebuttal (Part 3/3)**
>
> **Q4. Extended baselines**
>
> Thanks for providing these important baselines:
> - **Least-to-Most** [1]: Ask LLMs to break down a complex problem into a series of simpler subproblems and then solve them in sequence.
> - **SelfCheck** [2]: Utilize LLMs to check their own outputs and use the results of these checks to improve question-answering performance by conducting weighted voting on multiple solutions to the question.
> Here we adopt voting with 4 candidates.
>
> As the reviewer suggested, we also compared a lightweight version of SelfCheck, which utilize the same PRM in our framework as the checker.
>
> | **LLM** | **Method** | **MATH-500** | **GSM8K** | **GPQA** | **StrategyQA** | **Average** |
> |:---:|:---:|:---:|:---:|:---:|:---:|:---:|
> | Qwen2.5-Instruct-7B | DirectQA | 74.96±1.37 | 90.95±1.06 | 31.16±0.23 | 68.53±0.53 | 66.40±0.46 |
> |  | Least-to-Most | 73.28±0.84 | 89.10±0.82 | 31.79±0.54 | 73.60±0.64 | 66.94±0.25 |
> |  | SelfCheck | 74.36±1.60 | 90.24±1.09 | 33.84±0.61 | 72.87±0.77 | 67.83±0.35 |
> |  | SelfCheck-PRM | 75.96±1.20 | 89.78±0.89 | 33.71±0.51 | 72.84±0.64 | 68.07±0.20 |
> |  | **RLoT (ours)** | **76.88±1.35** | **92.95±0.63** | **44.78±0.71** | **79.56±0.64** | **73.54±0.18** |
>
> Both Least-to-Most and SelfCheck generally outperform DirectQA, with SelfCheck showing stronger improvements on average.
> Also, replacing the LLM self-checker with a specialized PRM yields a slight performance gain, confirming that the PRM provides useful signals.
> Combining the advantage of PRM and our design on task-adaptive logical structures, RLoT outperforms these baselines.
> These results clarify that our performance gains are not merely due to accessing a PRM.
> Instead, RL Navigator leverages these signals to dynamically construct task-adaptive logical structures, offering reasoning capabilities beyond simple one-shot verification.
>
> [1] Zhou et al. Least-to-most prompting enables complex reasoning in large language models. ICLR, 2023.\
> [2] Miao et al. Selfcheck: Using llms to zero-shot check their own step-by-step reasoning. ICLR, 2024.
>
>
> **Q5. Comparison with fixed logical schedules**
>
> We thank the reviewer for suggesting this point to separate the gains of the RL policy from the benefit of simply utilizing logic blocks.
>
> While strictly equalizing token counts is challenging due to the generative nature of LLMs, we conducted controlled experiments by fixing the number of logic blocks to ensure a relatively fair comparison of reasoning depth:
> - **Repeated Refine**: Repeat Refine until reaching the answer.
> - **Repeated Refine fixed-3**: Repeat Refine for three times and then terminate.
> - **Fixed Logic Sequence**: Decompose $\rightarrow$ Reason $\rightarrow$ Refine.
> - **RLoT fixed-3**: Use RL Navigator to output three blocks and then terminate. If the Navigator decides to terminate early, it is forced to pad the sequence with "Refine" actions until the length requirement is met.
>
> | **LLM** | **Method** | **GSM8K** | **GPQA** | **StrategyQA** | **Average** |
> |:---:|:---:|:---:|:---:|:---:|:---:|
> | Qwen2.5-Instruct-7B | DirectQA | 91.58 | 31.25 | 68.85 | 63.89 |
> |  | Repeated refine | 87.72 | 32.37 | 72.34 | 64.14 |
> |  | Repeated refine fixed-3 | 86.43 | 30.80 | 61.14 | 59.46 |
> |  | Decompose-Reason-Refine | 88.38 | 36.16 | 71.13 | 65.22 |
> |  | RLoT fixed-3 | 91.74 | 41.29 | 76.42 | 69.82 |
> |  | **RLoT (ours)** | **92.87** | **44.64** | **79.04** | **72.18** |
>
> The results confirm that while the mere presence of logic blocks offers some benefit, the adaptability of the RL policy is the primary driver of performance.
> Specifically, the Fixed Logic Sequence outperforms DirectQA, indicating that structured reasoning is inherently helpful.
> However, blindly applying blocks can be detrimental: the Fixed Repeated Refine setting led to a performance drop, as forcing the model to refine an already correct answer often resulted in "over-correction" and hallucinated errors.
> Furthermore, while the Constrained RLoT outperformed the fixed schedules, it still lagged behind the original RLoT.
> This demonstrates that the performance improvement is not just due to "having blocks," but stems from the RL Navigator's ability to dynamically select the most appropriate logic block for the specific state and to decide the moment to terminate the reasoning process.
>
> \
> \
> Thanks again for your valuable suggestions.
> We have made brief revisions to the manuscript accordingly and will continue to work to incorporate the above discussions and improvements into the final version.

---

### Official Review · Reviewer_oJop · 2025-11-02

**Soundness:** 2
**Presentation:** 3
**Contribution:** 3
**Rating:** 4
**Confidence:** 3

**Summary:**

The authors hypothesise that inference-time reasoning frameworks such as Chain of Thought (CoT), Tree of Thoughts (ToT), and Graph of Thoughts (GoT) are overly rigid and fail to adapt dynamically to different reasoning problems. To address this, they propose a reinforcement learning–based controller (“navigator”) that, given the current reasoning state representation, selects among five “logic blocks”: Reason, Decompose, Debate, Refine, or Terminate.

Reasoning is formalised as a Markov Decision Process (MDP) where states are defined by the LLM’s self-evaluation, actions correspond to logic blocks, and rewards are provided via a process reward model (PRM). The navigator is trained using a Double-Dueling DQN and subsequently used at inference time to guide the LLM’s reasoning adaptively.

**Strengths:**

- The idea of viewing reasoning as an MDP with composable cognitive actions is original and conceptually appealing. The method is explained clearly.

- The proposed navigator has fewer than 3K parameters, making it lightweight, computationally inexpensive, and compatible with frozen LLMs.

- The authors experimented across reasoning, math, STEM, and commonsense domains, and also covered both small- and medium-sized models.

- The authors show that their method can perform cross-model and cross-task generalisation.

**Weaknesses:**

- There’s no ablation on reward signals or PRM accuracy, so it’s unclear whether improvements come from RL navigation or simply from PRM-guided prompting.
- Some parts of the paper are not detailed enough, for example, section 4.5. How many instances were considered? The authors do not justify why Double-Dueling DQN is used. There is no exploration of policy stability, convergence, or sample efficiency.
- The paper majorly lacks qualitative analysis. The paper presents only anecdotal examples of reasoning patterns (“Reason–Refine–Debate”) and no failure-case analysis, error typology, or robustness study. We don’t know when or why RLoT fails.
- Minor point: The paper over-relies on the appendix for critical methodological and experimental details. The main text should ideally be more self-contained.

**Questions:**

- What happens if the self-evaluation state is noisy or replaced with random values — does the navigator still help?

---

> ### Author Response · Authors · 2025-11-21
> **Rebuttal (Part 1/3)**
>
> We sincerely thank the reviewer for the insightful suggestions.
> Below we provide a point-to-point response with additional experimental results and discussions.
>
> **Q1. Ablation on reward signals from the PRM**
>
> We thank the reviewer for raising this important point regarding the source of our performance gains.
>
> **First**, PRM is not used at inference first, we would like to clarify this potential misunderstanding.
> Unlike search-based methods (e.g., ToT or rStar) that rely on PRM to filter or select paths during inference, RLoT utilizes PRM solely as a reward provider during the training of the Navigator.
> Once trained, the Navigator operates independently at inference time without any access to the PRM.
> Therefore, the performance improvements are directly attributable to the learned policy of the RL Navigator in generating task-adaptive logical structures, rather than real-time PRM-guided prompting.
>
> **Second**, we have conducted extensive ablations in Appendix H.3 regarding reward types and PRM quality.
>
> | **LLM** | **PRM** | **GSM8K** | **GPQA** | **StrategyQA** | **Average** |
> |:---:|:---:|:---:|:---:|:---:|:---:|
> | Qwen2.5-Instruct-7B | None (DirectQA) | 91.58 | 31.25 | 68.85 | 63.89 |
> |  | ORM | 91.96 | 41.52 | 73.94 | 69.14 |
> |  | MathShepherd-Mistral-7B + disturb | 92.49 | 42.41 | 77.15 | 70.68 |
> |  | MathShepherd-Mistral-7B | 92.87 | 44.64 | 79.04 | 72.18 |
> |  | Qwen2.5-Math-PRM-7B | 93.10 | 45.31 | 80.06 | 72.82 |
>
> The ORM-trained Navigator still achieves significant gains over the baseline, demonstrating that the RL framework effectively learns reasoning strategies even without dense process supervision.
> Also, we tested the Navigator's sensitivity to PRM quality by introducing Gaussian noise and by using a stronger PRM (Qwen2.5-Math-PRM-7B).
> The results show that while higher-quality PRM yields better performance, our method is robust even with a noisy PRM which can still provide sufficient directional guidance.
>
> These results confirm that while a good PRM helps, the core improvement stems from the RL Navigator's ability to learn and generalize efficient reasoning structures.
>
>
> **Q2. Detailed information on RL training**
>
> Thanks for this question.
>
> **First**, the core contribution of this work lies in the architecture of the inference-time Navigator and its MDP formulation, rather than the specific RL algorithm used for training.
> Our framework is algorithm-agnostic, allowing for the employment of arbitrary off-the-shelf RL algorithms with discrete action space.
>
> Therefore, Double-Dueling DQN is just a standard RL algorithm for the specific implementation, and it is not very proper to include it in the Method.
> Now we have revised Mehtod Section 3.3 to clarify the algorithm-agnostic feature, and we newly added Section 4.2 for training details, where we explicitly justify the choice Double-Dueling DQN: it is a commonly used RL algorithm, which improves the stability during the training process.
>
> **Second**, we have added experimental analyses regarding the RL training process in Appendix A.2.
> We investigate policy stability by comparing Double-Dueling DQN (DDDQN) with a standard Double DQN (DDQN), and we extended the training duration to 5,000 episodes.
>
> | **LLM** | **Method** | **Olympic math** |  | **Elementary math** |  | **STEM** |  | **Commonsense** | **Average** |
> |---|---|---|---|---|---|---|---|---|---|
> |  |  | AIME24 | AMC23 | MATH | GSM8K | GPQA | MMLU-STEM | StrategyQA |  |
> | Qwen2.5-14B-Instruct | Direct QA | 13.33±0.00 | 56.50±1.22 | 78.55±0.40 | 93.95±1.04 | 36.74±0.63 | 85.01±0.23 | 72.90±0.80 | 62.43±0.18 |
> |  | Zero-shot CoT | 16.67±2.98 | 62.50±1.58 | 78.59±0.39 | 94.22±0.65 | 37.99±0.55 | 85.54±0.62 | 76.30±0.78 | 64.55±0.70 |
> |  | Few-shot CoT | 6.67±0.00 | 55.50±1.87 | 79.91±0.41 | 94.39±0.91 | 45.18±0.85 | 85.18±0.47 | 78.05±0.46 | 63.55±0.19 |
> |  | CoT-SC | 6.67±0.00 | 45.50±1.00 | 80.08±0.47 | 93.66±0.35 | 45.13±0.74 | 86.48±0.45 | 79.91±1.03 | 62.49±0.25 |
> |  | ToT | 10.00±0.00 | 55.50±1.87 | 79.71±0.43 | 93.83±1.44 | 45.36±0.71 | 86.51±0.37 | 78.49±0.87 | 64.20±0.31 |
> |  | RLoT (DDDQN-3k) | **24.67±1.63** | **65.50±1.87** | **80.47±0.39** | 94.24±1.01 | 51.43±0.94 | 89.10±0.45 | 80.47±0.62 | **69.41±0.35** |
> |  | RLoT (DDDQN-5k) | 22.67±2.49 | 64.00±1.22 | 80.32±0.44 | 94.33±0.52 | **52.14±0.67** | 89.01±0.42 | 80.73±1.17 | 69.03±0.42 |
> |  | RLoT (DDQN-5k) | 24.00±2.49 | 65.00±1.58 | 80.32±0.36 | **94.47±0.91** | 51.61±1.14 | **89.42±0.40** | **80.73±1.00** | 69.36±0.32 |
>
> As shown in Figure 4 and the above table, while the Double DQN eventually converges to a comparable performance level, it exhibits slightly decreased training stability compared to the Double-Dueling architecture.
> This empirical evidence supports our decision to use Double-Dueling DQN as the default algorithm.
> The results also show that the model essentially reaches convergence around 3,000 episodes with only marginal gains thereafter, confirming our setting of training episodes.

---

> ### Author Response · Authors · 2025-11-21
> **Rebuttal (Part 2/3)**
>
> **Q3. Noise in the self-evaluation state**
>
> We appreciate this insightful question regarding the robustness of the Navigator and its dependency on the state representation.
> To address this, we have included a detailed quantitative analysis in Appendix F.2, where we deliberately introduced synthetic noise into the self-evaluation state vectors during inference.
>
> | **LLM** | **State** | **GSM8K** | **GPQA** | **StrategyQA** | **Average** |
> |:---:|:---:|:---:|:---:|:---:|:---:|
> | Qwen2.5-Instruct-7B | DirectQA | 91.58 | 31.25 | 68.85 | 63.89 |
> |  | Random | 89.76 | 28.13 | 65.50 | 61.13 |
> |  | 50% noise | 91.74 | 34.82 | 68.41 | 64.99 |
> |  | 30% noise | 92.27 | 43.75 | 71.32 | 69.11 |
> |  | **Normal** | **92.87** | **44.64** | **79.04** | **72.18** |
>
> When we replaced the self-evaluation state with random values, the performance dropped significantly to an average of 61.13%, which is even lower than the Direct QA baseline.
> This confirms that the Navigator actively utilizes the specific semantic information in the state vector to make decisions, rather than simply learning a fixed, state-agnostic policy or sequence.
>
> However, the framework exhibits robustness to noise.
> Even when 50% of the state bits were randomly flipped, RLoT maintained an average accuracy of 64.99%, which is still comparable to the Direct QA baseline.
> As the noise decreased to 30%, the performance recovered rapidly to 69.11%, showing clear improvements over the baseline.
>
> Meanwhile, to minimize potential noise inherent in self-evaluation, we deliberately designed the evaluation criteria as a binary classification task rather than a multi-grade continuous scoring.
> Continuous or fine-grained scoring (e.g., 1-10 scales) requires precise calibration, which is challenging for LLMs.
> In contrast, a binary standard simplifies the decision boundary, significantly reducing the cognitive load on the LLM.
> This design choice makes the evaluation process more robust, ensuring that the state vector remains a stable and reliable signal for the navigator even with less capable LLMs.
>
>
> **Q4. Failure-case and robustness analysis**
>
> We thank the reviewer for emphasizing the need for deeper qualitative analysis.
>
> **First**, in the reasoning pattern analysis, we analyzed the entire test sets of the respective benchmarks.
> We have explicitly listed these numbers in the revised Table 5:
> - MATH: 5,000 instances
> - GPQA: 448 instances
> - StrategyQA: 687 instances
>
> **Second**, we have added Appendix B.2, which provides a detailed error typology and failure analysis. We categorize common failures into two primary modes:
> - **Type 1**: Navigator Input Errors (Self-Evaluation Noise).
> As illustrated in Figure 6, this occurs when the LLM hallucinates during the self-evaluation phase (e.g., marking an incorrect step as "True").
> This noise misleads the Navigator in selecting action, propagating the error.
> - **Type 2**: Navigator Output Errors (Action Selection Error).
> As illustrated in Figure 7, this occurs when the Navigator selects a suboptimal action despite receiving correct state information.
> For instance, the Navigator might trigger an unnecessary "Refine" on a correct step, causing the LLM to "over-correct" and introduce new errors.
>
> While these failure modes exist, our extensive quantitative results demonstrate that RLoT achieves performance gains over baselines, indicating that these failure cases represent a minority of the inference trajectories.
> In our design, the RL training process effectively optimizes the navigator to minimize action selection errors, and the structured self-evaluation prompt is designed to mitigate state noise errors as much as possible.
> It is also an important future direction to develop safeguard mechanisms that help the framework to avoid common failure situations.

---

> ### Author Response · Authors · 2025-11-21
> **Rebuttal (Part 3/3)**
>
> **Q5. Self-contain of the main text**
>
> We thank the reviewer for pointing this out.
> We apologize for the inconvenience caused by the strict space limitations of the initial submission, which forced us to move some details to the Appendix.
>
> Taking advantage of the one-page extension allowed during the rebuttal phase, we have improved the self-containment of the main text by incorporating critical methodological and experimental details:
> - **RL Training Details (Section 4.2)**: We have added a subsection to detail the specific RL algorithm used and included the learning curves to demonstrate the convergence of our training process.
> - **Ablation Studies (Section 4.7)**: We have integrated the key ablation study regarding our core logical blocks design into the to better justify the effectiveness of each module.
>
> We are committed to further optimizing the content and structure in the final version to ensure the main text provides a comprehensive and self-contained presentation of our method.
>
> \
> \
> Thanks again for your valuable suggestions.
> We have made brief revisions to the manuscript accordingly and will continue to work to incorporate the above discussions and improvements into the final version.

---

### Meta-Review · Area_Chair_Hb4B · 2025-12-09

**Summary:**

Reviewers were mostly positive in their initial reviews, with 3/4 voting on acceptance. Common concerns included insufficient number of ablations and lack of confidence intervals.

**Reviewer Concerns:**

Ablation studies to determine the source of performance improvement (oJop, HceF)
- Addressed by referring to Appendix H.3

Confidence intervals (TZiL, VvQn)
- Addressed by updating Table 2 with confidence intervals

No major concerns were unaddressed.

**Reviewer Scores:**

I think reviewer oJop would have acknowledged the new results and raised their score to 6.

Other reviewers were already positive, and their concerns were primarily about the method's design choices, or its performance improvement over baselines. The rebuttal does not resolve these concerns, and their (positive) scores would have remained unchanged.

---

### Decision · Program_Chairs · 2026-01-26

Accept (Poster)